# It Is Always the Same—A Complication Classification following Angular Stable Plating of Proximal Humeral Fractures

**DOI:** 10.3390/jcm12072556

**Published:** 2023-03-28

**Authors:** Georg Siebenbürger, Rouven Neudeck, Mark Philipp Daferner, Evi Fleischhacker, Wolfgang Böcker, Ben Ockert, Tobias Helfen

**Affiliations:** 1Department of Orthopaedics and Trauma Surgery, Musculoskeletal University Center Munich (MUM), University Hospital, LMU Munich, 81377 Munchen, Germany; 2Augenklinik, Städtisches Klinikum, 76133 Karlsruhe, Germany

**Keywords:** proximal humeral fracture, complications, complication classification, angular stable plating locking plate osteosynthesis, mid-term, functional outcome, constant score

## Abstract

Introduction: The aim of this study was to create a novel complication classification for osteosynthesis-related complications following angular stable plating of the proximal humerus subsuming the influence of these complications on clinical outcome in relation to fracture morphology and consequent revision strategies. A total of 1047 proximal humerus fractures with overall 193 osteosynthesis-associated complications (24.5%) were included. The following complication types could be clarified: complication Type 1 is defined by mild varus (<20°) or valgus displacement of the humeral head without resulting in a screw cutout through the humeral head cortex. Type 2a is defined by varus displacement (<20°) of the humeral head associated with screw cutout through the humeral head cortex. Type 2b complication is limited to displacement of the greater tuberosity, lesser tuberosity, or both tuberosities. Complication Type 2c is defined by severe varus dislocation (>20°) of the humeral head with screw cutout at the humeral head cortex. Complication Type 3 describes a displacement of the angular stable plate in the humeral shaft region with associated shaft-sided screw cutout, while the position of the humeral head remains static. Complication Type 4 is characterized by the occurrence of AVN with or without glenoidal affection (4a/b). Clinical outcome according to the constant score was mainly affected by type 2–4, leading to a deteriorated result. Depending on the type of complication, specific revision strategies can be considered. Additionally, more complex fracture patterns fostered the incidence of complications.

## 1. Introduction

Proximal humerus fractures are the third most common fractures in elderly patients (>60 years), with the incidence increasing overall, especially in multi-part fractures [1].

The increasing incidence is reported by Rupp et al. within the years 2009 to 2019 by 10% with a total of 61.606 proximal humerus fractures, which equates to an incidence of 90.8 per 100,000 inhabitants in Germany [2]. While the general consensus for nondisplaced proximal humerus fractures is conservative treatment with a sling and early physiotherapy [3], the treatment of displaced and unstable fractures remains open to debate [1]. However, the most established surgical procedure is open reduction and internal fixation by angular stable plating, with a complication rate of up to 30% [4,5,6,7,8]. Schuetze et al. recently published an overall complication rate of 21% in geriatric patients after minimally invasive locking plating [9].

Furthermore, published studies show a significantly higher complication rate and worse functional outcome following open reduction and internal fixation (ORIF) of the proximal humerus in elderly patients than reconstruction in younger patients [8,10,11,12,13,14]. A variety of complications are described in the literature leading to proximal humerus fracture sequelae after locking plate osteosynthesis: primary, secondary, osteosynthesis-associated, and non-osteosynthesis-associated complications. They all include postoperative infections, hematoma, nerve injury, rotator injury, frozen shoulder, impingement, displacement/fixation loss, screw cutout, implant failure due to fracture or screw loosening, and avascular necrosis [8,10]. In terms of frequency, secondary displacement, screw cutout, and AVN of the shoulder merit interest [15,16,17,18]. Going further, these are the most common complication patterns resulting in revision surgery [18]. Furthermore, there is a significant incidence increase for implant-associated complications in elderly patients, age >70 years, and not for fracture type, whereas for non-implant-associated complications, a significant incidence increase is found for fracture type severity and not for patient age [10]. There are several techniques described to reduce secondary displacement after angular stable plate osteosynthesis of proximal humeral fractures. Recently, Schuetze et al. described a significantly lower rate of implant-associated complications, especially regarding revision surgery after screw tip augmentation [9]. Comparable results could be demonstrated for strut grafts [19]. Nevertheless, the overall complication rate regarding osteosynthesis-related complications following angular stable plating of displaced proximal humeral fractures remains the most important problem.

The aim of this study is to assess the functional outcome and osteosynthesis-related complications of displaced fractures of the proximal humerus after surgical treatment with angular stable plate osteosynthesis in a large cohort. These complications will further be defined and radiologically analyzed, characterized, and classified. Furthermore, we tried to investigate the impact of these complications on clinical outcome and relation to fracture morphology, as well as potential revision strategies.

## 2. Materials and Methods

### 2.1. Selection Criteria

Between February 2002 and December 2014, a collective of 1031 patients with 1047 displaced proximal humerus fractures were included in this retrospective cohort study at the Musculoskeletal University Center Munich of the Ludwig-Maximilians-University, Munich. In some cases, the patients sustained fractures of both sides at different time points. The inclusion criteria for this study were consecutive patients (>18 years) treated by angular stable plate osteosynthesis of a displaced proximal humeral fracture. The surgical indication for ORIF was made in all patients based on Neer’s criteria (angulation >45°, fracture displacement >1 cm).

Patients with following parameters were excluded:
Polytraumatized;diagnosed stroke or dementia;open and pathologic fractures;primary nerve/vessel injuries;primary screw misalignment;primary unsuccessful surgical reduction (>5° head–shaft displacement, cranialization of the greater tuberosity of >5 mm and valgus head–shaft alignment >150° or <110° head shaft angle according to Schnetzke et al., 2016) [20].

The indication for surgery was based on Neer criteria (>45° angulation or >1cm separation between major fracture segments). The surgical technique was performed in a standardized manner as previously described by Ockert et al. [11]. The rotator cuff was intraoperatively evaluated, and in cases of rupture, if possible, fixated to the plate with sutures during the operative procedure.

Clinical total follow-up regarding Constant score was available in 557 (53.2%) cases, with a mean follow-up of 4.0 ± 2.7 years and median of 3.3 years, and the mean age of the 557 cases was 65.3 ± 14.5 years. More patients could be completely followed up radiologically without a complete clinical outcome. In total, 787 cases (75.2%) were available for radiological outcome/complication analysis with different follow-up times. The mean follow-up was 7.3 ± 14.0 months, with a median of 2.8 months. The mean age of the 787 patients was 66.5 ± 15.6 years, and the gender distribution showed 532 (67.6%) women and 255 (32.4%) men. For detailed follow-up, see Figure 1.

### 2.2. Radiographic Assessment

The criteria for the presence of osteosynthesis-associated complications by means of the postoperative follow-up radiographs have been defined as secondary displacement >10° in the varus or valgus direction, offset of >5 mm, screw cutout, and AVN. Exclusion criteria were insufficient anatomic reduction in the first postoperative radiograph (>10° axial deviation, >5 mm displacement) and primary screw malposition in the first postoperative radiograph within the first days after surgery [15].

This assessment was applied to all consecutive x-radiograph series, which in addition to identifying complication patterns, allowed an evaluation of time of onset and progression over time with differentiation of progression and constancy. Furthermore, these complication patterns are completed by the collected clinical results and by the analysis of the documentation of performed revisions with corresponding procedure selection. Based on this collection, different complication types have been defined. The extent of varus displacement is determined using a standard radiographic measurement program (see Figure 2). The reference value for an anatomical humeral head–shaft angle corresponds to 135° [21].

### 2.3. Statistical Analysis

For data collection and analysis, the software SPSS (IBM Corp. Released 2020. IBM SPSS Statistics, Version 21.0. Armon, NY, USA: IBM Corp) and Excel (Microsoft, released 2021, Microsoft Excel) were used.

The test procedure used to compare the dependent variable of two groups was the T-test for independent, normally distributed samples (effect size measure Cohen’s d and r) and, if the normal distribution requirement was not met, the Mann–Whitney U-test (effect size measure r). Differences with a *p*-value < 0.05 were defined as significantly different. For the difference measurement of the dependent variable between more than two independent groups, the one-factor ANOVA (effect size measure Eta-squared η² and Cohen’s f) was used if the normal distribution assumption was met, and the Kruskal–Wallis test (effect size measure r) was used if the normal distribution assumption was violated. Subsequent post-hoc tests (LSD and Dunn–Bonferroni) ensured pairwise comparisons. Simple regression analysis was used to test for a relationship between two metric-scaled variables (effect strength measure Cohen’s f²). Pearson’s correlation test was used to determine the linear, bivariate relationship. The Pearson Chi-square test was used to test for a relationship between categorical variables (effect size measure Cramer’s V). In contrast, Fisher’s exact test was used for expected frequencies <5. Due to the size of the cross-tabulations, the subsequent Bonferroni correction was used to determine the causality of this correlation.

## 3. Results

### 3.1. Radiographic Evaluation

The 1047 initial trauma X-rays series of the shoulder (a.p. and scapula Y view) have been analyzed, and the distribution according to the Neer classification shows and are illustrated in Figure 3:308 (29.6%) 2-part fractures;471 (45.3%) 3-part fractures;173 (16.6%) 4-part fractures;36 (3.5%) head-split fractures;52 (5.0%) dislocated fractures.

**Figure 3 jcm-12-02556-f003:**
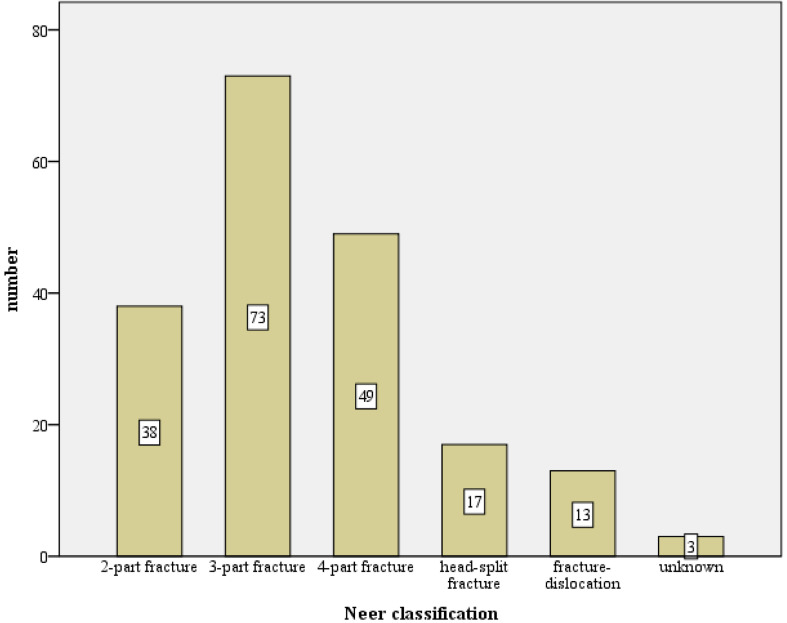
Absolute fracture distribution of the total collective, based on the Neer classification, total number n = 1047 (X-axis: Neer classification, Y-axis: number).

According to AO classification, (see Figure 4)

20 (1.9%) A1 fractures;112 (10.78%) A2 fractures;161 (15.5%) A3 fractures;252 (24.25%) B1 fractures;214 (20.6%) B2 fractures;32 (3.08%) B3 fractures;65 (6.26%) C1 fractures;152 (14.63%) C2 fractures and;31 (3.0%) C3.

**Figure 4 jcm-12-02556-f004:**
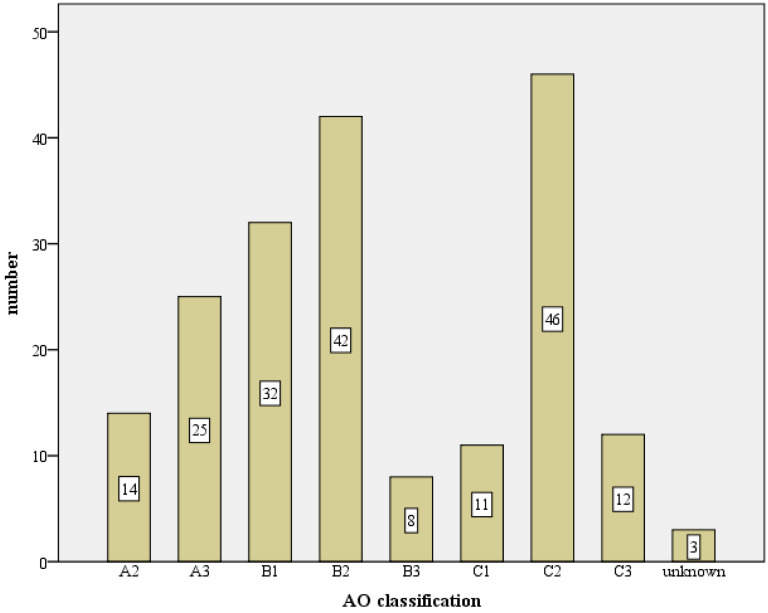
Absolute fracture distribution of the total collective, based on the AO-classification, total number n = 1047 (X-axis: AO-classification, Y-axis: number).

### 3.2. Complication Types

Osteosynthesis-related complications were divided into the following classification system in accordance with the 2020 published study by Fleischhacker et al., which demonstrated the relation of varus displacement to functional outcome following ORIF of proximal humeral fractures. Functional clinical outcome is inferior in cases with humeral head- shaft angle displacement of 10 to 20° varus in comparison to an anatomical head–shaft angle of 135°. A varus malposition/displacement of >20° resulted in even worse functional outcome as well as screw cutout. Based on the observations and analysis of the postoperative x-rays during the follow-up, different types of complication could be defined [24]. 

#### 3.2.1. Complication Type 1

Complication type 1 is defined by a secondary mild, stable varus or valgus displacement (<20°) of the humeral head without resulting in a screw cutout through the humeral head cortex (see Figure 5, Figure 6 and Figure 7).

#### 3.2.2. Complication Type 2

Complication Type 2 has three different subgroups. As a common feature, Type 2a and 2c are accompanied by displacement of the humeral head associated with a screw cutout through the humeral head cortex. Type 2b is defined as displacement of the greater or lesser tuberosity or both.

##### Complication Type 2a

Complication Type 2a is defined by varus displacement (<20°) of the humeral head associated with screw cutout through the humeral head cortex. The head sintering is considered relatively stable and non-progressive (Figure 8, Figure 9 and Figure 10).

##### Complication Type 2b

Complication Type 2b is limited to displacement of the greater tuberosity, the lesser tuberosity, or both tuberosities (Figure 11 and Figure 12).

##### Complication Type 2c

Type 2c is defined by a severe varus displacement (>20°) of the humeral head, which is associated with a screw cutout at the humeral head in parallel to Type 2a. In contrast to Type 2a, this varus displacement is characterized by an unstable osteosynthesis situation and progression (Figure 13, Figure 14 and Figure 15).

#### 3.2.3. Complication Type 3

This complication describes a displacement of the angular stable plate osteosynthesis with screw cutout/fracture avulsion in the humeral shaft region, while the humeral head position remains intact (Figure 16 and Figure 17).

#### 3.2.4. Complication Type 4

Complication Type 4 is characterized by the occurrence of humeral head necrosis, despite initially correct osteosynthesis. In divergence to the previously described complication types, it must not be the result of secondary displacement. Type 4 is divided into two subgroups, representing the possible involvement of damage to the glenoid articular surface. 

##### Complication Type 4a

Complication Type 4a is based on the presence of AVN of the humeral head without concomitant destruction of the glenoid articular surface due to an associated screw cutout through the humeral head cortex (Figure 18 and Figure 19).

##### Complication Type 4b

Complication Type 4b is also based on the presence of an AVN of the humeral head combined with a destruction of the glenoid articular surface due to the associated screw cutout through the humeral head cortex (Figure 20 and Figure 21).

Table 1 shows an overview of all complication types with sample X-rays and pattern images.

### 3.3. Results of the Complication Types

The descriptive distribution of complication types is shown in Table 2. A total of 193 complications occurred, resulting in a complication rate of 24.5%.

Comparatively, the collective without complications (n = 594) was younger, with a mean age of 64.8 ± 16.0 years. They were distributed among 400 women (67.3%) and 194 men (32.7%).

Functional outcomes, based on the CS, nCS, and %CS of each complication type and the complication collective; the cohort without complications; and the case cohort in total for contrast, are shown in Table 3. For the complication types, the following functional outcomes resulted based on the scored Constant score:

Complication Type 1 = moderate to good, Complication Type 2a = moderate, Complication Type 2b = poor, Complication Type 2c = poor, Complication Type 3 = poor to moderate, Complication Type 4a = poor, Complication Type 4b = poor.

When looking at the mean CS between the CC (n = 440, mean 72.1, SD 19.0) and the respective complication Type 1–4b (mean and SD, see Table 3), there was continuously a statistically significant difference, which was based on the following levels:Complication Type 1 (n = 37) was on average 7.4 points lower compared to the cohort without complications (CC) (95% CI [1.1, 13.8]), t (475) = 2.305, *p* = 0.022, d = 0.4, r = 0.1.Complication Type 2a (n = 29) was on average 13.4 points lower than CC (95% CI [6.3, 20.5]), t (467) = 3.699, *p* < 0.001, d = 0.7, r = 0.2.Complication Type 2b (n = 5) was on average 38.9 points lower than CC (95% CI [22.2, 55.7]), t (443) = 4.571, *p* < 0.001, d = 2.1, r = 0.2.Complication Type 2c (n = 9) was on average 32.8 points lower than CC (95% CI [20.3, 45.3]), t (447) = 5.138, *p* < 0.001, d = 1.7, r = 0.2.Complication Type 3 (n = 7) was on average 19.4 points lower than CC (95% CI [5.3, 33.6]), t (445) = 2.699, *p* = 0.007, d = 1.0, r = 0.1.Complication Type 4a (n = 23) was on average 25.7 points lower than CC (95% CI [17.8, 33.7]), t (461) = 6.364, *p* < 0.001, d = 1.4, r = 0.3.Complication Type 4b (n = 7) was on average 25.8 points lower than CC (95% CI [11.7, 40.0]), t (445) = 3.585, *p* < 0.001, d = 1.4, r = 0.2.

When analyzing the mean nCS between the CC (n = 440, mean 84.7, SD 21.9) and the complication Type 1–4b (mean and SD, see Table 3), there was also a statistically significant difference, which was based on the following levels:Complication Type 1 (n = 37) was on average 8.5 points lower compared to the cohort without complications (CC) (95% CI [1.1, 15.8]), t (475) = 2.270, *p* = 0.024, d = 0.4, r = 0.1Complication Type 2a (n = 29) was on average 14.7 points lower compared to the CC (95% CI [6.4, 22.9]), t (467) = 3.496, *p* < 0.001, d = 0.7, r = 0.2.Complication Type 2b (n = 5) was on average 44.7 points lower compared to the CC (95% CI [25.3, 64.0]), t (443) = 4.543, *p* < 0.001, d = 2.0, r = 0.2.Complication Type 2c (n = 9) was on average 36.7 points lower compared to the CC (95% CI [22.2, 51.2]), t (447) = 4.977, *p* < 0.001, d = 1.7, r = 0.2.Complication Type 3 (n = 7) was on average 21.0 points lower compared to the CC (95% CI [4.7, 37.3]), t (445) = 2.529, *p* = 0.012, d = 1.0, r = 0.1.Complication Type 4a (n = 23) was on average 30.5 points lower compared to the CC (95% CI [21.3, 39.7]), t (461) = 6.527, *p* < 0.001, d = 1.4, r = 0.3.Complication Type 4b (n = 7) was on average 31.2 points lower compared to the CC (95% CI [14.9, 47.6]), t (445) = 3.758, *p* < 0.001, d = 1.4, r = 0.2.

Assessment of the mean %CS between CC (n = 411, MW 87.2, median 90.1, SD 27.9) and the respective complication Type 1–4b (mean and SD see Table 3) showed the following result:Complication Type 1 (n = 31) was on average 2.7 percentage points lower than CC (95% CI [−7.3, 12.7]), but thus showed no statistically significant difference, t (440) = 0.532, *p* = 0.595. There was a statistically significant difference between complication Type 2a (n = 28, median = 76.4) and CC, U (411, 28) = 3796, z = −3.015, *p* = 0.003, r = 0.14.Complication Type 2b (n = 5) was on average 45.2 percentage points lower than CC (95% CI [20.5, 69.8]), thus represented a statistically significant difference, t (414) = 3.599, *p* < 0.001, d = 1.6, r = 0.2. There was a statistically significant difference between complication Type 2c (n = 9, median = 60.0) and CC, U (411, 9) = 725, z = −3.123, *p* = 0.002, r = 0.15.Complication Type 3 (n = 6) was on average 13.1 percentage points lower than CC (95% CI [−9.4, 35.7]), but showed no statistically significant difference, t (415) = 1.145, *p* = 0.253.Complication Type 4a (n = 21) was on average 29.9 percentage points lower than CC (95% CI [17.7, 42.0]), reflected a statistically significant difference, t (430) = 4.830, *p* < 0.001, d = 1.1, r = 0.2.Complication Type 4b (n = 6) was on average 34.3 percentage points lower than CC (95% CI [11.8, 56.8]), which reflected a statistically significant difference, t (415) = 2.996, *p* = 0.003, d = 1.2, r = 0.2.

A one-factor ANOVA was calculated to state the differences in the functional outcome of CS and nCS, depending on the different complication types. Here, the CS was statistically significantly different for the variance of complication types, F (6, 110) = 6.235, *p* < 0.001, η² = 0.25, f = 0.6. Similarly, the nCS showed a statistically significant difference for the same variance of complication types, F (6, 110) = 5.805, *p* < 0.001, η² = 0.24, f = 0.6. The Kruskal-Wallis test confirmed that %CS was also statistically significantly different between complication types, Chi-square (6) = 33.969, *p* < 0.001.

Using LSD-Post-hoc tests, there were significant differences for CS between the following complication types:

Complication Type 1:-31.5 points higher than complication Type 2b (95% CI [15.5, 47.5]), *p* < 0.001.-25.4 points higher than complication Type 2c (95% CI [12.9, 37.8]), *p* < 0.001.-18.3 points higher than complication Type 4a (95% CI [9.4, 27.2]), *p* < 0.001.-18.4 points higher than complication Type 4b (95 CI [4.6, 32.2]), *p* = 0.009.

Complication Type 2a:-25.5 points higher than complication Type 2b (95% CI [9.3, 41.8]), *p* = 0.002.-19.4 points higher than complication Type 2c (95% CI [6.6, 32.2]), *p* = 0.003.-12.3 points higher than complication Type 4a (95% CI [3.0, 21.7]), *p* = 0.01.

Using LSD post-hoc tests, there were significant differences for the nCS between the following complication types:

Complication Type 1:
-36.2 p. higher than Complication Type 2b (95% CI [16.8, 55.6]), *p* < 0.001.-28.2 p. higher than Complication Type 2c (95% CI [13.1, 43.4]), *p* < 0.001.-22.0 p. higher than Complication Type 4a (95% CI [11.2, 32.8]), *p* < 0.001.-22.8 p. higher than Complication Type 4b (95% CI [6.0, 40.0]), *p* = 0.008.

Complication Type 2a:-30.0 p. higher than Complication Type 2b (95% CI [10.3, 49.7]), *p* = 0.003.-22.0 p. higher than Complication Type 2c (95% CI [6.5, 37.6]), *p* = 0.006.-15.8 p. higher than Complication Type 4a (95% CI [4.4, 27.2]), *p* = 0.007.

The Dunn–Bonferroni tests showed that only complication Types 1 and 2b (z = 3.602, *p* = 0.007, r = 0.6), complication Types 1 and 4a (z = 4.433, *p* < 0.001, r = 0.6), and complication Types 1 and 4b (z = 3.274, *p* = 0.022, r = 0.5) differed significantly for the %CS.

The remaining group comparisons of complication types showed no significant difference in terms of functional outcome in post-hoc analysis based on CS, nCS, and %CS; even though descriptive differences of the respective means were presented.

The CS, nCS, and %CS of the collective without complications and of each complication type are shown below (see Figure 22, Figure 23 and Figure 24).

### 3.4. Complication Management

In the postoperative follow-up, all revision-surgery was recorded. These varied depending on the complication type. Revision surgery included implant removal (IR), which was further subdivided into early IR (<9 months) based on timing; screw replacement or screw removal; reosteosynthesis; humeral head/tuberosity resection; and procedure change to hemiarthroplasty, reverse arthroplasty, or CRIF, such as intramedullary nailing. A total of 136 revisions were performed (with some multiple revisions per case, 15% of the total radiologically examined collective, 61% of the osteosynthesis-associated complication collective). The dark-shaded fields in Table 4 accentuate the dominant revision type depending on the complication type. Only complication Type 1 showed a predominance in the absence of revision. If revision was considered necessary, it was mainly performed by IR. For complication Type 2a, early IR and screw replacement/screw removal were the dominant revision procedures. Complication Type 2b showed a more heterogeneous distribution pattern. Typical revisions were reosteosynthesis and procedure change to hemiarthroplasty or reverse arthroplasty. For complication Type 2c, conversion to arthroplasty (reverse > hemiarthroplasty) represented the superior revision procedure, followed by reosteosynthesis and screw change/removal. For complication Type 3, reosteosynthesis was the determining revision procedure. The complication of AVN (complication Type 4) mostly required conversion to arthroplasty; in addition, an IR was frequently performed.

It was shown that the complication types and the need for revision (yes/no) were related (Chi-Square (5) = 43.69, *p* < 0.001, n = 193). This association could be classified as medium (Cramer’s V = 0.48). Post-hoc analysis using Bonferroni correction indicated that the expected and observed values of type 1 and type 4 regarding revision yes/no were responsible for this statistically significant association.

### 3.5. Association with Fracture Classification

The absolute values of complication cases in association with the respective fracture type, according to the Neer and AO classifications, can be seen in Figure 25 and Figure 26.

The percentage frequencies of developing an osteosynthesis-associated complication in association with the Neer classification are shown in Table 5. The risk was highest for head split fractures with 51.5%, followed by 4-part fractures with 35.8% and dislocation fractures with 32.5%. Figure 25 and Figure 26 show a comparison of the number of cases with and without complications in relation to the fracture type (Neer and AO classifications).

By assigning the individual complication types to the original fracture types (Neer and AO classification), the numbers listed in Table 6 could be determined. The fields with a dark background indicate the dominant fracture patterns.

It was shown that both Neer fracture types (Chi-square (4) = 32.9, *p* < 0.001, n = 783, Cramer’s V = 0.21) and AO fracture types (Chi-square (2) = 14.2, *p* < 0.001, n = 782, Cramer’s V = 0.14) were related to the manifestation of a complication (yes/no). Post-hoc analysis using Bonferroni correction indicated that the expected and observed values in 4-part fracture, head-split fracture, and Type C fracture regarding complication yes/no were responsible for this statistically significant association.

The analysis of the fracture types and the need for revision (yes/no) shows that there was a statistically significant association according to both Neer (Fisher’s exact test, value = 22.66, *p* < 0.001) and AO (Chi-square (2) = 9.96, *p* = 0.007, n = 782, Cramer’s V = 0.11). The decisive factors were 4-part fracture, head-split fracture, and Type C fracture with regard to revision yes/no, which was demonstrated by post-hoc analysis.

## 4. Discussion

### 4.1. Overview of Complications after Locking Plate Osteosynthesis

Complication rates after angular stable plate osteosynthesis vary in the literature and depend on the respective complication definition and complication classification. Often, these overall complication rates include primary complications such as primary screw cutout [9,11].

The separate subdivision of osteosynthesis-associated and fracture-associated complications was introduced in the present study because a clear separation often cannot be presented. An evaluation of classic surgical risks/complications such as hematoma, wound infection, and nerve lesion was omitted due to the research question [11,15].

In summary, common complications after locking plate osteosynthesis are secondary varus displacement, secondary screw cutout, and humeral head necrosis [1,6]. For example, Ockert et al. described a secondary loss of fixation in 21%, a screw cutout in 9%, and an AVN in 5% in a cohort of 43 patients after locking plate osteosynthesis of the proximal humerus [8]. Probabilities of occurrence in this regard were reported by Owsley et al. (2008) [18], with 25% for secondary varus displacement, 23% for secondary screw cutout, and 4% for humeral head necrosis [11]. In the present work, 193 of the 787 fractures evaluated were associated with a secondary osteosynthesis-associated complication, resulting in a complication rate of 24.5%. This was composed of varus displacement at 15.2%, screw cutout at 8.4%, humeral head necrosis at 5.3%, isolated humeral shaft displacement at 2.3%, and tuberosity displacement at 1.7%. The rate turns out to be lower than in the comparable literature, corresponding to the decrease in the complication rate [6,8,11,21].

The complication collective (CS 54.5 ± 19.0 p., nCS 64.5 ± 22.9 p., %CS 71.2 ± 25.0%) was functionally significantly worse (*p* < 0.001) compared to the collective without complications (CS 72.1 ± 19.0 p., nCS 84.7 ± 21.9 p., %CS 87.2 ± 27.9%). Consequently, the complications in aggregate do justice to the statement of having a worse functional outcome. Significantly worse functional outcome of the complication collective compared to the non-complications was also confirmed by other published studies [8].

### 4.2. Description of Types of Osteosynthesis-Associated Complications

An essential aspect of the present work was to anatomically describe and classify individual types of complications by radiological diagnostics and to investigate their functional outcome and the need for revision surgery.

The aspect of varus displacement is represented by complication Types 1, 2a, and 2c. Complication Type 2b represents the isolated displacement of the greater tuberosity, the lesser tuberosity, or the combination of both tuberosities, resulting in a rotator-cuff insufficiency. Complication Type 3 is characterized by secondary displacement of the humeral shaft area, while the humeral head position simultaneously proves to be stable and intact. The aspect of humeral head necrosis is described by complication Type 4, despite primary correct osteosynthesis and absence of secondary displacement. The difference between Type 4a and 4b is the additional destruction of the glenoid articular surface by an associated screw cutout in the case of Type 4b.

The degree of secondary varus displacement in this study ranges from 15.6° ± 6.1° for Type 1 to 16.3° ± 3.9° for Type 2a to 27.4° ± 4.6° for Type 2c. The functional result was congruent in the inverse direction. Functionally, a moderate to good result resulted for Type 1 (CS 64.7 ± 16.7 p., nCS 76.2 ± 20.3 p., %CS 84.5 ± 18.0%), a moderate result for Type 2a (CS 58.7 ± 17.5 p., nCS 70.0 ± 21.6 p., %CS 77.5 ± 24.6%), and a poor result for Type 2c (CS 39.3 ± 17.1 p., nCS 48.0 ± 21.1 p., %CS 65.3 ± 28.5%), so that for the first two complication types the outcome can be pronounced a satisfactory functional outcome. Complication Type 2b, as a reflection of tuberosity displacement, represented a poor functional outcome (CS 33.2 ± 13.0 p., nCS 40.0 ± 16.6 p., %CS 42.0 ± 20.2%). Complication Type 3, reflecting humeral shaft displacement, showed poor to moderate functional outcome (CS 52.7 ± 9.4 p., nCS 63.7 ± 12.2 p., %CS 74.1 ± 20.5%). The functional outcome of complication Type 4a (CS 46.4 ± 17.2 p., nCS 54.2 ± 20.4 p., %CS 57.3 ± 20.4%) and 4b (CS 46.3 ± 14.0 p., nCS 53.4 ± 14.9 p., %CS 52.9 ± 15.1%) resulting from AVN could be classified as poor. All types of complications showed a significantly worse score for the CS and nCS than the collective without complications (*p* < 0.05), so they prove the definition as a complication.

A comparison with other recent published studies is difficult due to the divergent classification of complications. However, individual aspects of the respective studies are listed below.

Solberg et al. (2009) also calculated the functionality by means of CS of individual complication groups [25]. Therefore, patients with humeral screw cutout achieved a CS of 67.8 ± 9.6 p., with a loss of fixation and secondary hemiarthroplasty a CS of 47.5 ± 3.5 p., and with AVN a CS of 62.5 ± 4.6 p. Based on the negative correlation of initial varus displacement and final CS, patients with initial varus <5° achieved a CS of 76 p. and with 5–20° a CS of 66 p., with a further varus displacement tendency of ≤20° a CS of 71.4 p. and of >20° a CS of 47 p. [11]. The CS values we observed are comparatively worse (see Table 3).

Brunner et al. (2009) reported a functional outcome one year postoperatively after angular stable plating. In this study, severe varus (CCD angle <115°) resulted in a mean CS of 60 p. and a %CS of 73%, whereas minor varus displacement (CCD angle 115–124°) resulted in a mean CS of 70 p. and a %CS of 83% [10]. Type 1, as we defined it, is functionally comparable to the low varus displacement value. Type 2a (low varus) and 2c (high varus) show us comparatively worse functional values. However, the different presentation of varus displacement as well as the additional differentiation of the cutout must be taken into account. The CS for operated patients with subsequent AVN of 48.7 p. determined by Greiner et al. (2009) coincides with the value for complication Type 4 [26].

Therefore, the new complication classification has a special position in the current literature with regard to functional outcome of individual complication types after angular stable plate osteosynthesis for proximal humerus fracture [5,7,10,11,13,14,16,22,23,27].

### 4.3. Association of Initial Fracture Type with Functional Postoperative Outcome and Complication

The fracture morphological distribution of 1047 fractures of the present work, as shown in Figure 3 and Figure 4, is overall comparable with the literature data [9,10,12,13,15]. While good functional results are obtained in the literature for 2- and 3-part fractures with angular stable plating, only a moderate result occurs for 4-part fractures (CS 67.7 p. and 67.6 p.) [15,16]. In contrast, a comparable functional outcome (CS 64.6 p., nCS 75.8 p., %CS 77.6%) after such a fracture can be reported in this study. In the literature, the dependence of functional outcome and fracture type is often presented [8,15,16,28].

The 193 fractures that resulted in one of the defined complication types were distributed as 38% 3-part fractures, 26% 4-part fractures, 20% 2-part fractures, 9% head-split fractures, and 7% dislocation fractures (see Figure 25) or as 43% Type B fractures, 36% Type C fractures, and 21% Type A fractures (see Figure 26). We observed a significant correlation for fracture types according to Neer and AO and the manifestation of a complication (*p* < 0.001) as well as need for revision surgery (Neer *p* < 0.001, AO *p* = 0.007). This was related to 4-part fractures, head-split fractures, and Type C fractures. Individual distribution patterns can be observed (see Table 6). Therefore, complication Types 1 and 2a were similarly and most frequent in 3-part fractures. Complication Type 2b arose mostly from a 4-part fracture or a Type C fracture. Interestingly, complication Type 2c showed a clustered occurrence of 2- and 3-part fractures, while 4-part fractures and Type C fractures were less frequent as the initial fracture pattern. One explanation for these frequencies could be inadequate medial support and anatomic reduction, which consequently led to loss of fixation with severe varus displacement despite mild fracture complexity.

Agudelo et al. (2007) investigated the associations of fixation loss and could not find a correlation with fracture type. Instead, they named malreduction in varus position <120° as a significant factor [6]. Owsley et al. (2008) also demonstrated no significant association of varus displacement and screw cutout with fracture type [18]. Complication Type 3 was primarily represented by 2- and 3-part fractures.

Based on the complex fracture patterns, it must be summarized that 51.5% of head-split fractures, 35.8% of 4-part fractures, and 32.5% of dislocated fractures developed a complication. An association of poor shoulder function and fracture complexity seems understandable, given the parallel congruence of fracture complexity and more severe complications.

Overall, it can be assumed that complex fractures, especially with questionable anatomical reduction, are more likely to lead to arthroplasty in the elderly patient collective in the future [29,30,31].

### 4.4. Limitation

The main limitation is certainly the retrospective study design (case series, evidence class III). The variability of the follow-up times (clinical and radiological) could not be listed, so that the information was given by means of mean and SD. Due to the size of the collective, there was also a greater variability with regard to the treating surgeons. The presence of only final functional values was also considered limiting. Consequently, no distinction was made between functional values before and after revision for the respective complication types.

## 5. Conclusions

It is always the same—this study shows a new classification system for postoperative osteosynthesis-associated complications following angular stable plating of proximal humeral fractures.

Summing up, all complication types show a poorer functional outcome in comparison to the collective without a complication. This work presents a classification of individual complication types and undertakes their analysis and anatomic descriptor in detail. Subsequent revision procedures and links to fracture type have also been integrated.

Overall, it can be assumed that complex fractures of the proximal humerus, which have been shown to increase the likelihood of a clinically relevant complication when using angular stable plate osteosynthesis, will in future tend to lead to arthroplasty in elderly patients in order to minimize complication rates. The correct indication is crucial. In addition to the possibility of typifying secondary, osteosynthesis-associated complications, the present study provides a complication-based overview as a clinical guide for angular stable plate osteosynthesis of the proximal humerus.

## Figures and Tables

**Figure 1 jcm-12-02556-f001:**
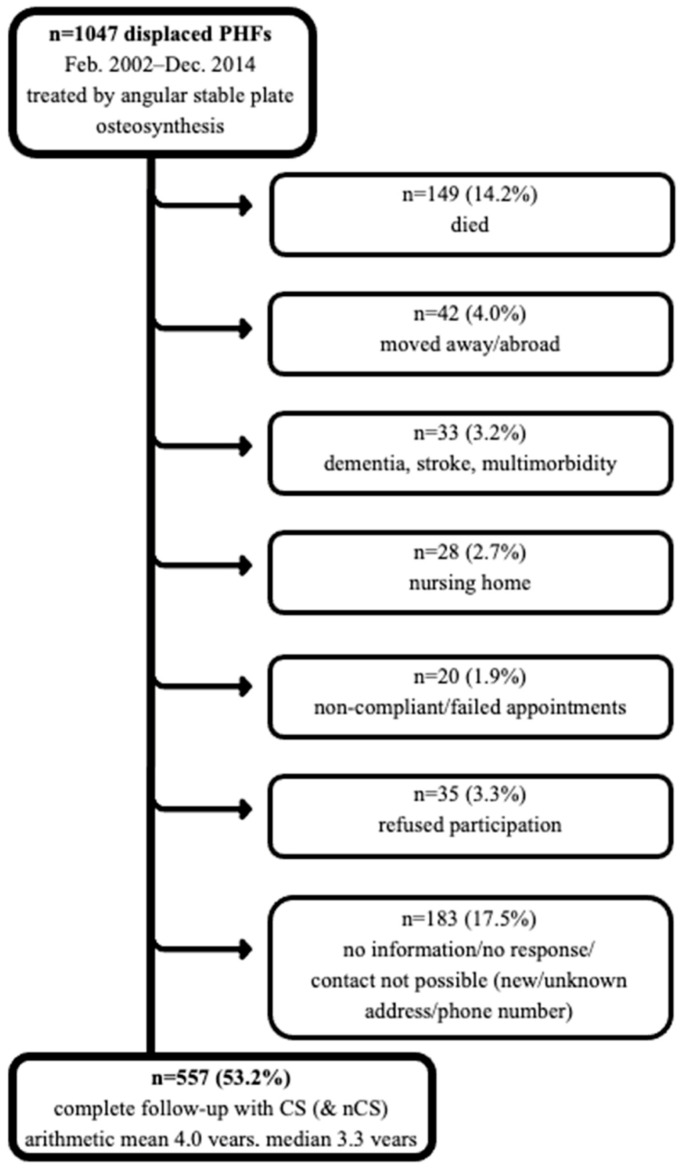
Flow chart of patients included at 4 years of follow-up.

**Figure 2 jcm-12-02556-f002:**
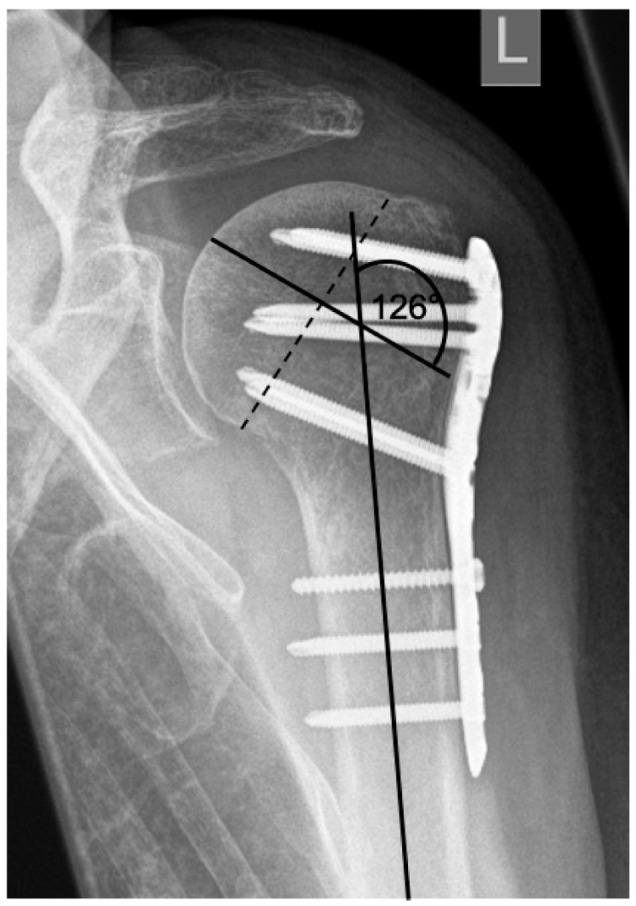
Calculation of the humeral head–shaft angle in the a.p. radiographic path by drawing three lines and the intersection of the line along the long axis of the humeral shaft with the perpendicular to the line of inferior + superior articular surface [22,23].

**Figure 5 jcm-12-02556-f005:**
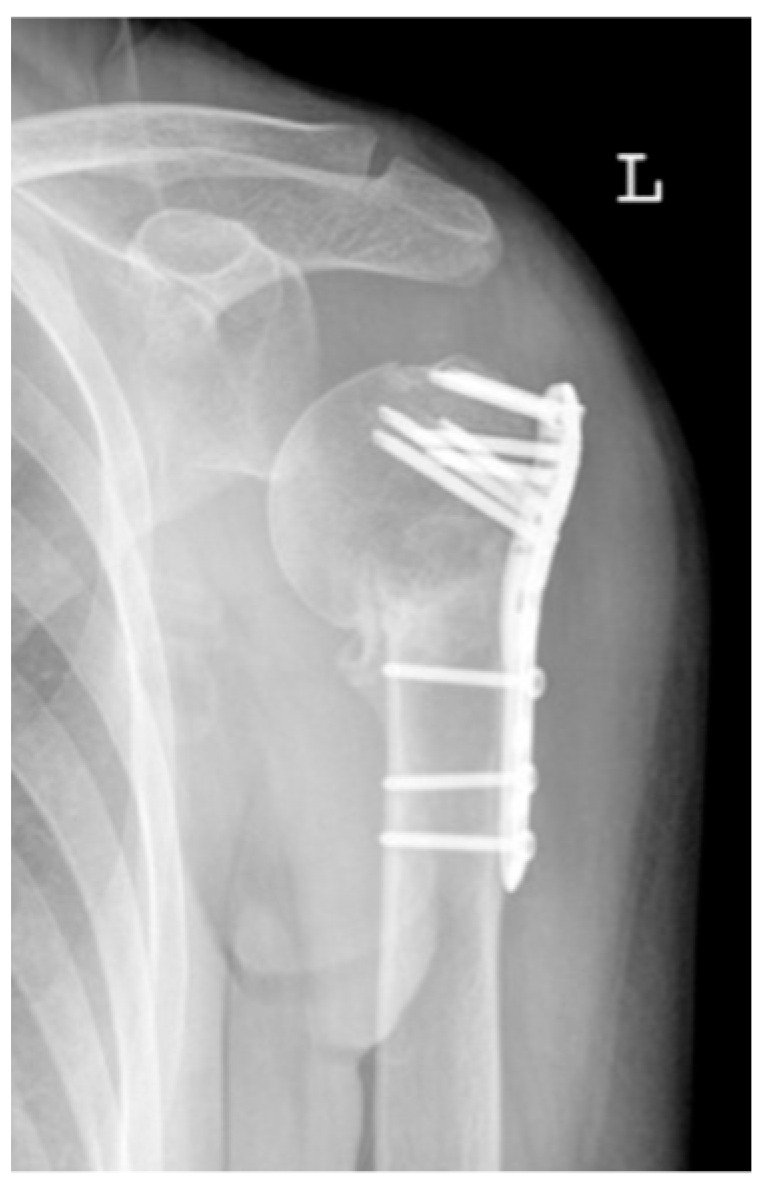
Complication Type 1. a.p. radiograph of the left shoulder after angular stable plate osteosynthesis illustrates mild varus displacement of the humeral head. The humeral head cortex remains intact, so there is no screw cutout.

**Figure 6 jcm-12-02556-f006:**
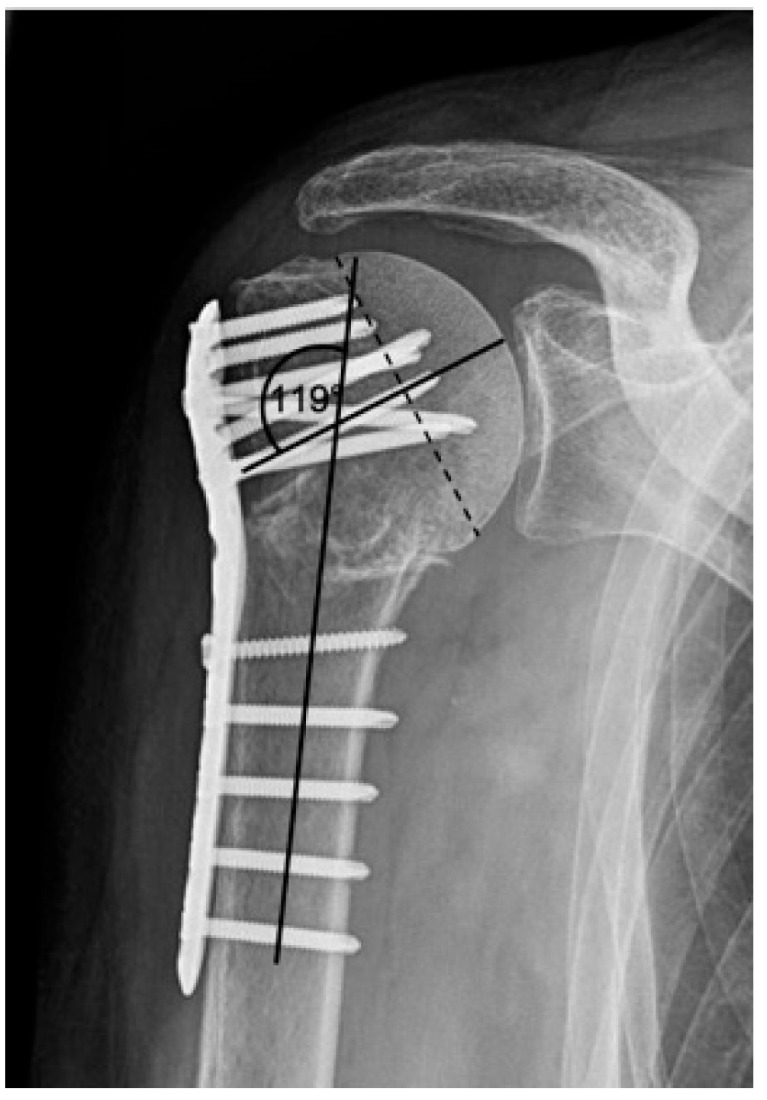
Complication Type 1. Middle: measurement of varus displacement on an a.p. radiograph of a right shoulder after angular stable plate osteosynthesis. The varus displacement is 16° relative to the anatomic CCD angle of 135°. No screw cutout results.

**Figure 7 jcm-12-02556-f007:**
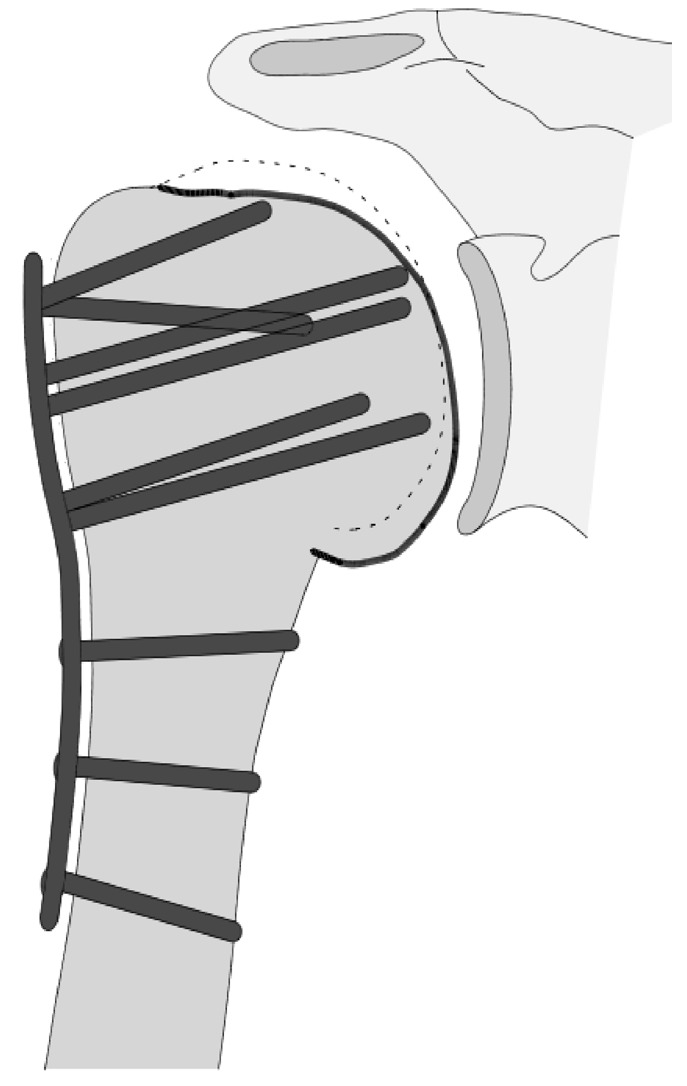
Complication Type 1. sample image. Mild varus displacement of the humeral head after angular stable plate osteosynthesis is shown by the transition of the dashed drawing (initially correct humeral head position) to the prominent black drawing.

**Figure 8 jcm-12-02556-f008:**
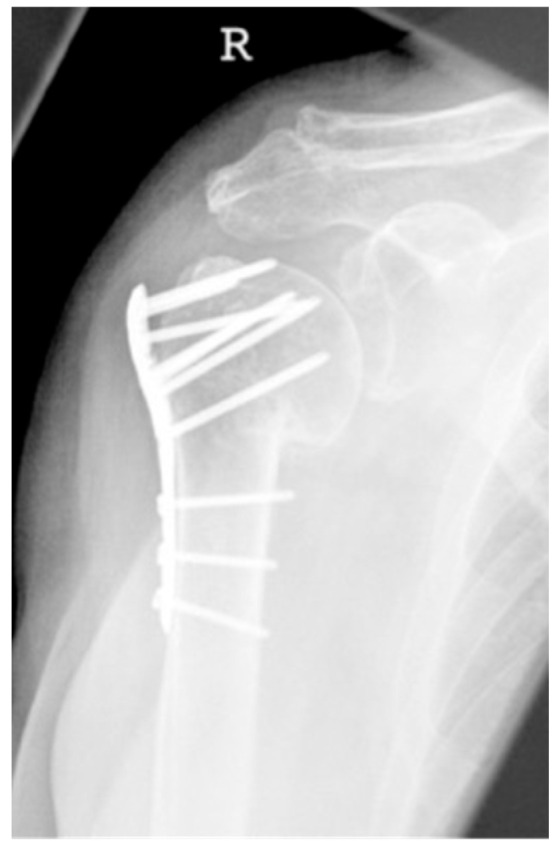
Complication Type 2a. a.p. radiograph of the right shoulder after angular stable plate osteosynthesis shows moderate varus displacement of the humeral head. The most cranial screw breaks through the humeral head cortex (screw cutout).

**Figure 9 jcm-12-02556-f009:**
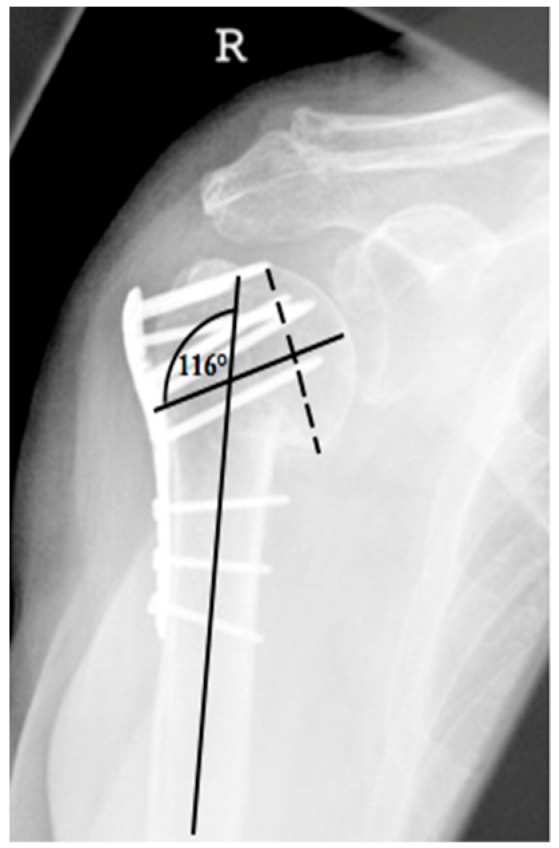
Complication Type 2a. CCD-relative varus displacement of 19°.

**Figure 10 jcm-12-02556-f010:**
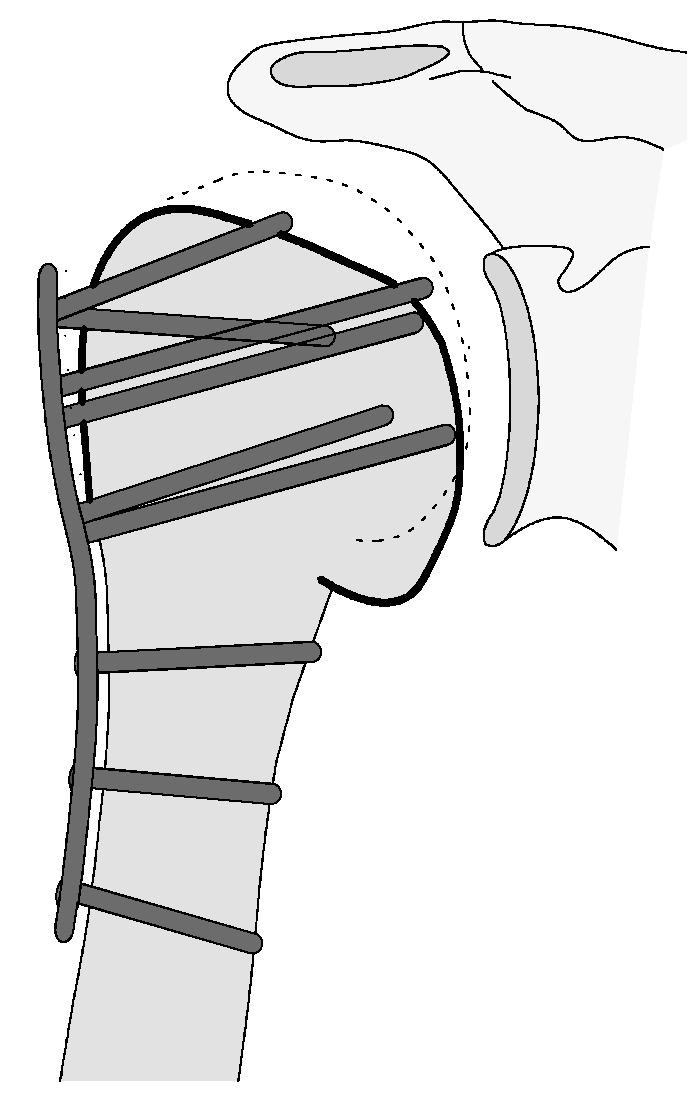
Complication Type 2a. Pattern image. Moderate grade varus displacement of the humeral head after angular stable plate osteosynthesis is illustrated by the transition of the dashed drawing (initially correct humeral head position) to the prominent black drawing with screw cutout (double).

**Figure 11 jcm-12-02556-f011:**
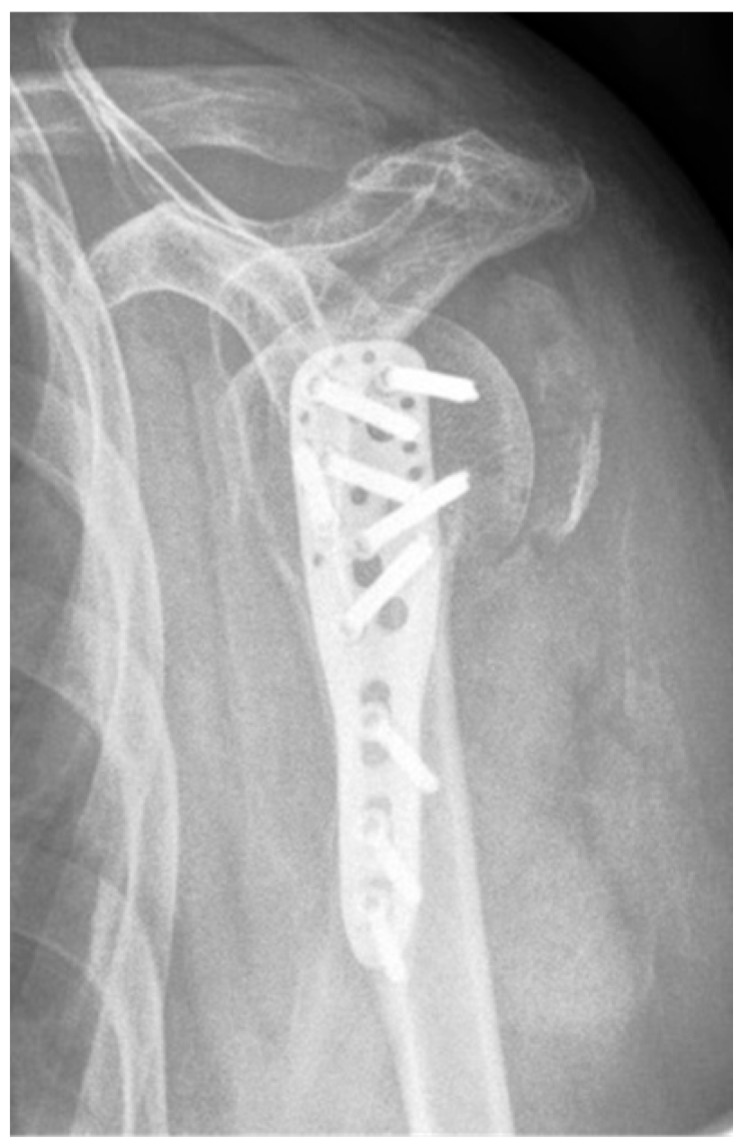
Complication Type 2b. Y-view radiograph of the left shoulder after angular stable plate osteosynthesis illustrates displacement of the greater tuberosity.

**Figure 12 jcm-12-02556-f012:**
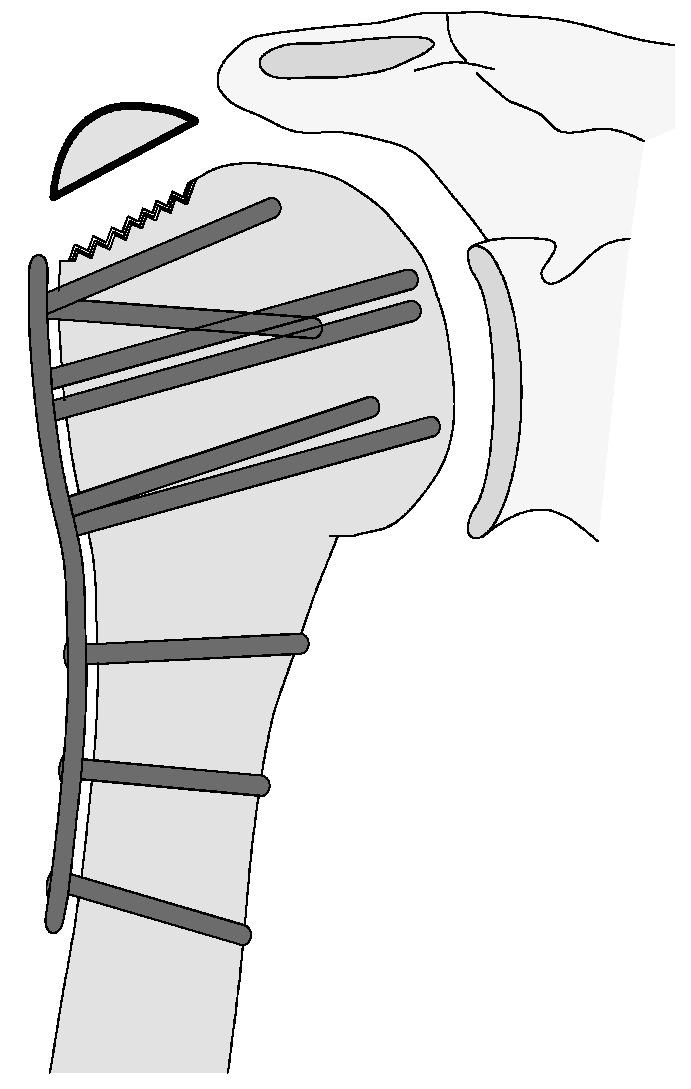
Complication Type 2b. Sample image. Following angular stable plate osteosynthesis, the displacement of the greater tuberosity is illustrated by the separation of the prominent black drawing from the serrated drawing (initial regular position).

**Figure 13 jcm-12-02556-f013:**
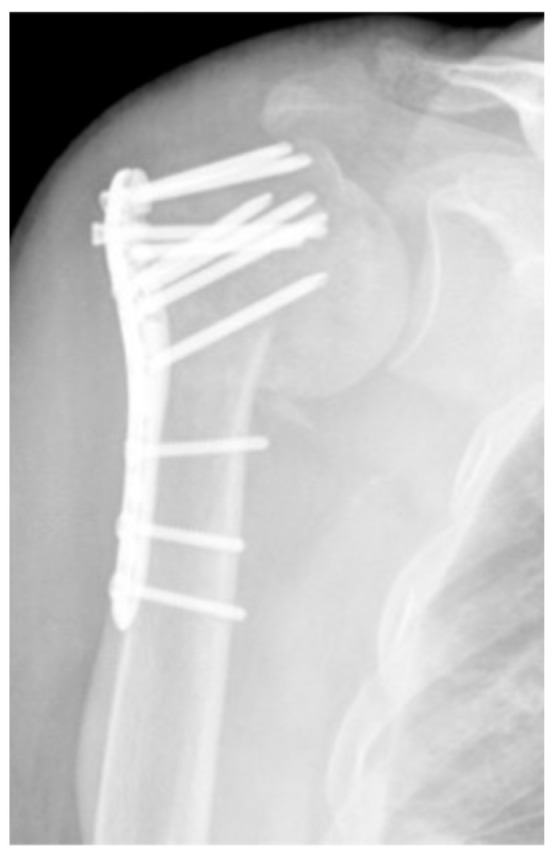
Complication Type 2c. a.p. radiograph of the right shoulder after angular stable plate osteosynthesis shows severe varus displacement of the humeral head. The most cranial screws penetrate the humeral head cortex, resulting in a screw cutout. The humeral head sintering appears unstable.

**Figure 14 jcm-12-02556-f014:**
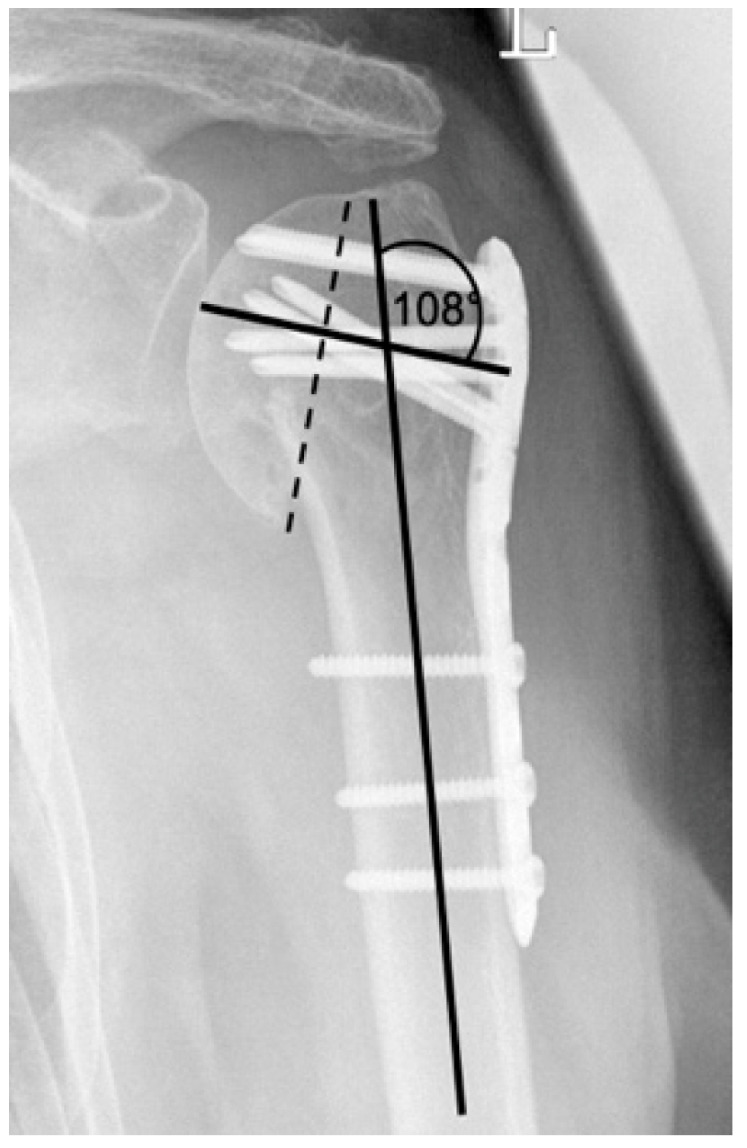
Complication Type 2c. Measurement of varus displacement on an a.p. radiograph of a left shoulder after angular stable plate osteosynthesis with same-sided complication. The cranial screws penetrate the cortical bone. Varus displacement is 27° relative to the anatomic CCD angle of 135°.

**Figure 15 jcm-12-02556-f015:**
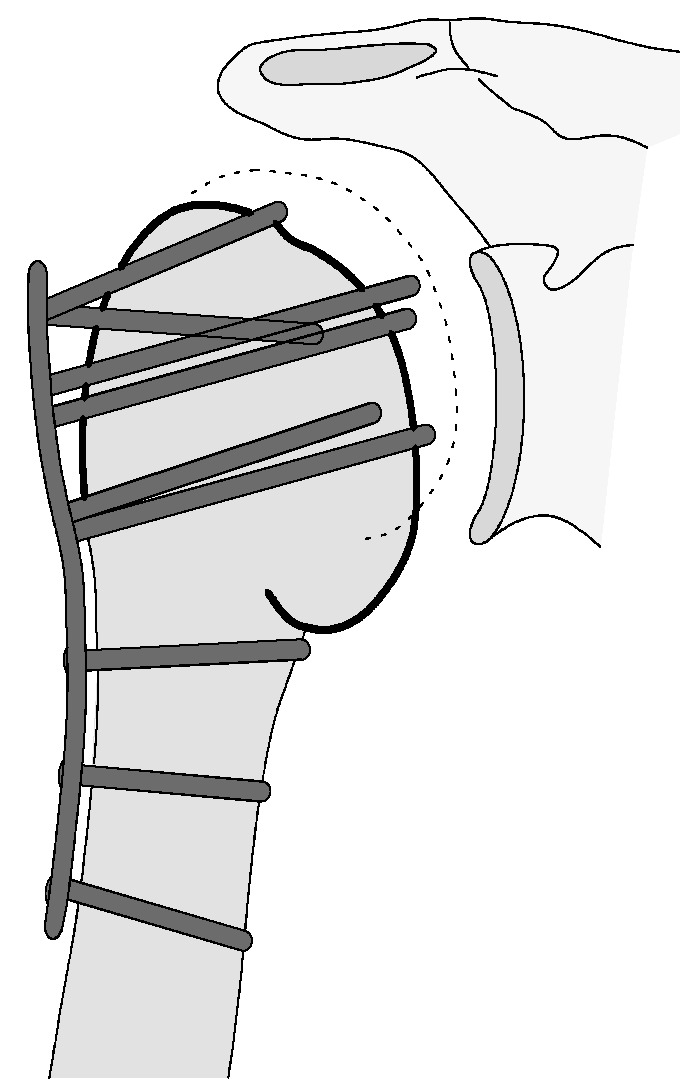
Complication Type 2c. Pattern image. Severe varus displacement and instability of the humeral head after angle-stable plate osteosynthesis is shown by the transition of the dashed drawing (initially correct humeral head position) to the prominent black drawing. This is followed by a cutout of four screws.

**Figure 16 jcm-12-02556-f016:**
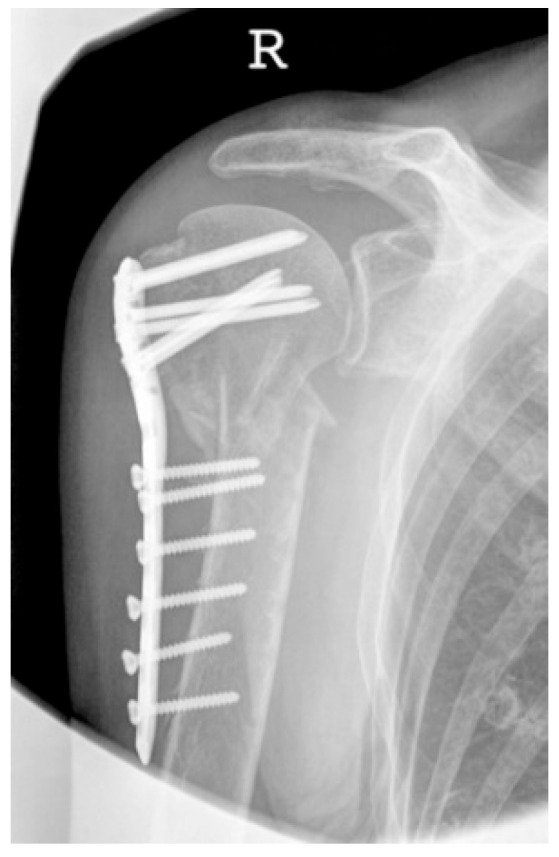
Complication Type 3. a.p. radiograph of the right shoulder after angular stable plate osteosynthesis shows a displacement of the humeral shaft area, so the fixation of the plate osteosynthesis is no longer guaranteed. In contrast, the humeral head position is regular.

**Figure 17 jcm-12-02556-f017:**
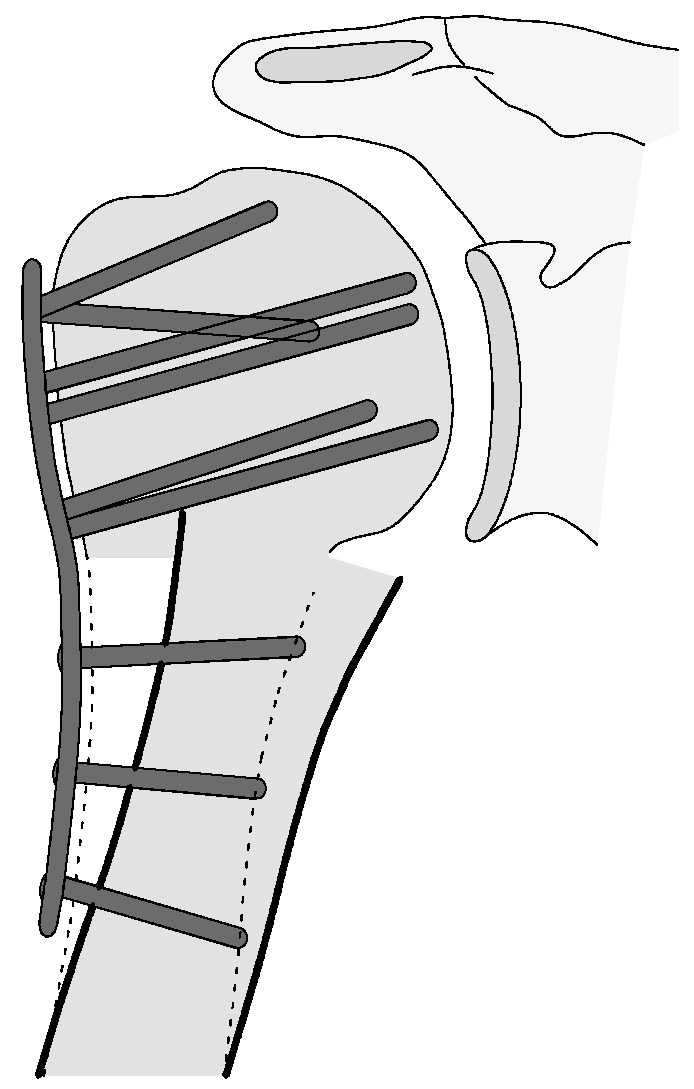
Complication Type 3. Pattern image. The displacement of the humeral shaft after angle-stable plate osteosynthesis is shown by the transition of the dashed drawing (initially correct humeral shaft position) to the prominent black drawing.

**Figure 18 jcm-12-02556-f018:**
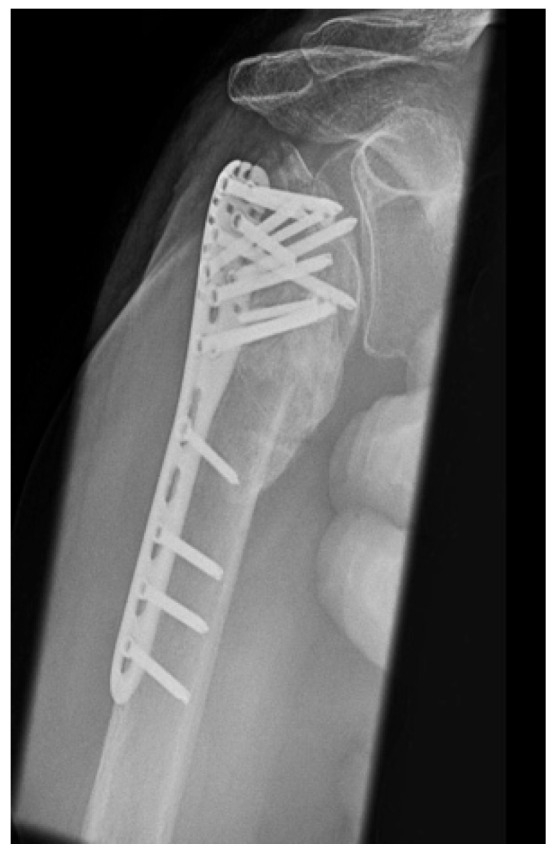
Complication Type 4a. a.p. radiograph of the right shoulder after angular stable plate osteosynthesis shows necrosis of the humeral head. The glenoid articular surface is still intact.

**Figure 19 jcm-12-02556-f019:**
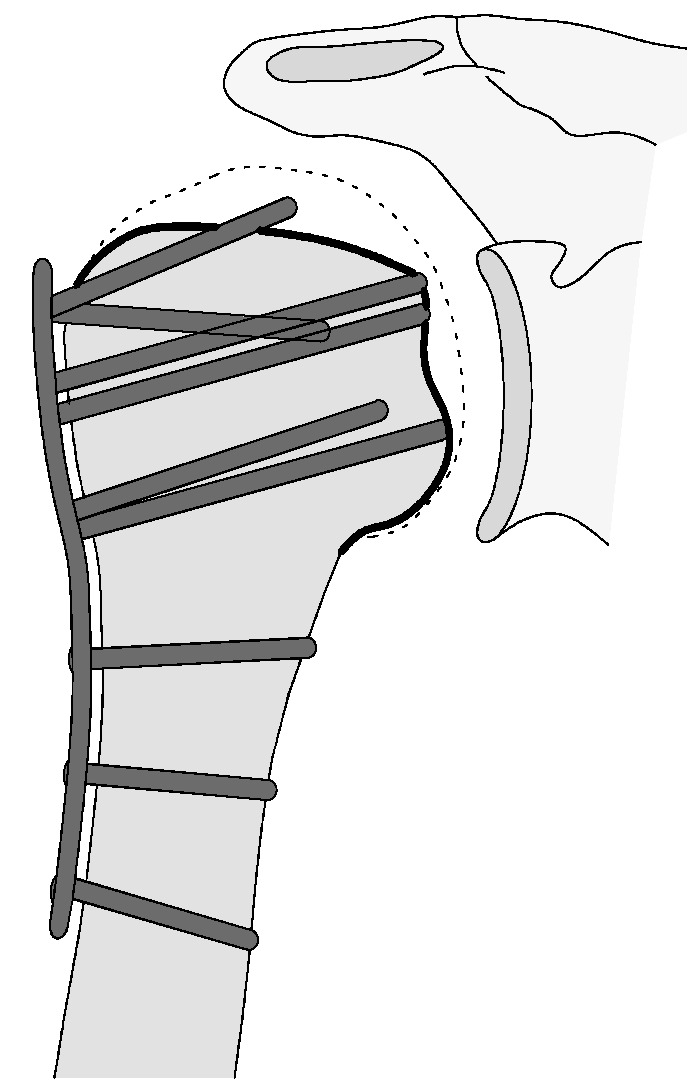
Complication Type 4a. Sample image. Humeral head necrosis after angle-stable plate osteosynthesis is depicted by the transition of the dashed drawing (regular humeral head anatomy) to the inhomogeneous, prominent black drawing. A cutout of the most cranial screw results, but this does not cause damage to the glenoid articular surface.

**Figure 20 jcm-12-02556-f020:**
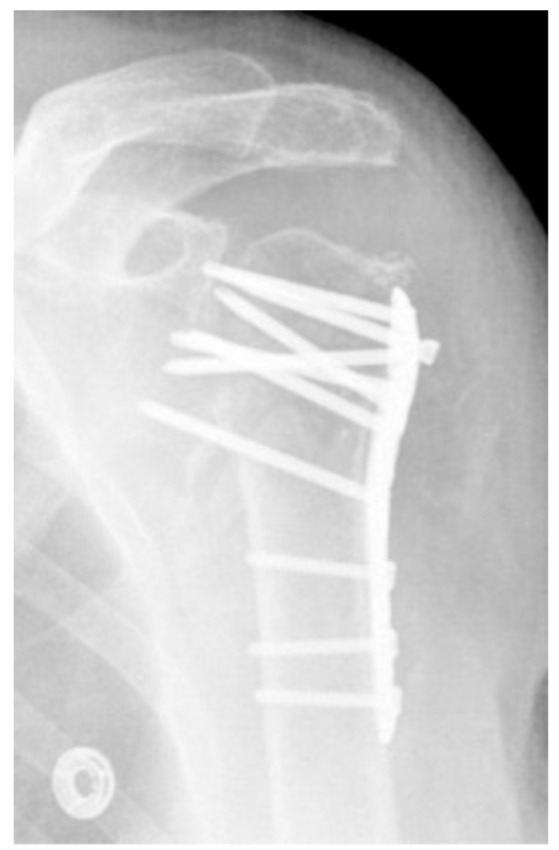
Complication Type 4b. a.p. radiograph of the left shoulder after angular stable plate osteosynthesis illustrates necrosis of the humeral head, implying cutout of multiple screws. This has resulted in arrosion of the glenoid articular surface.

**Figure 21 jcm-12-02556-f021:**
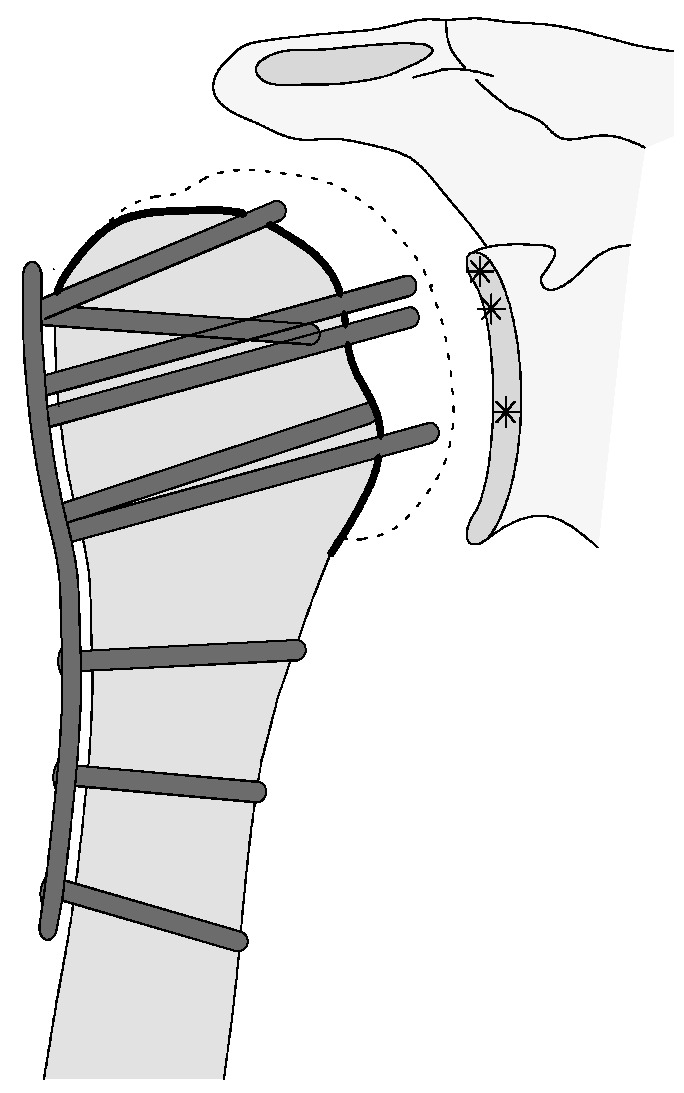
Complication Type 4b. Sample image. Humeral head necrosis after angular stable plate osteosynthesis is illustrated by the transition of the dashed drawing (regular humeral head anatomy) to the inhomogeneous, prominent black drawing. A multiple screw cutout results. The damage to the glenoid articular surface based on this is marked by the star symbols on the cavitas glenoidalis.

**Figure 22 jcm-12-02556-f022:**
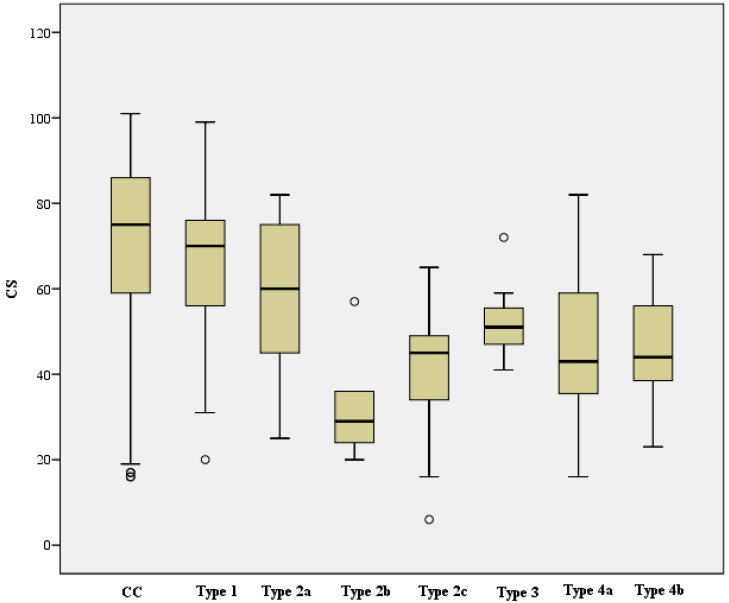
Comparison CS of collective without complications and complication types (X-axis: study collective, Y-axis: median, IQR, extreme values, outliers).

**Figure 23 jcm-12-02556-f023:**
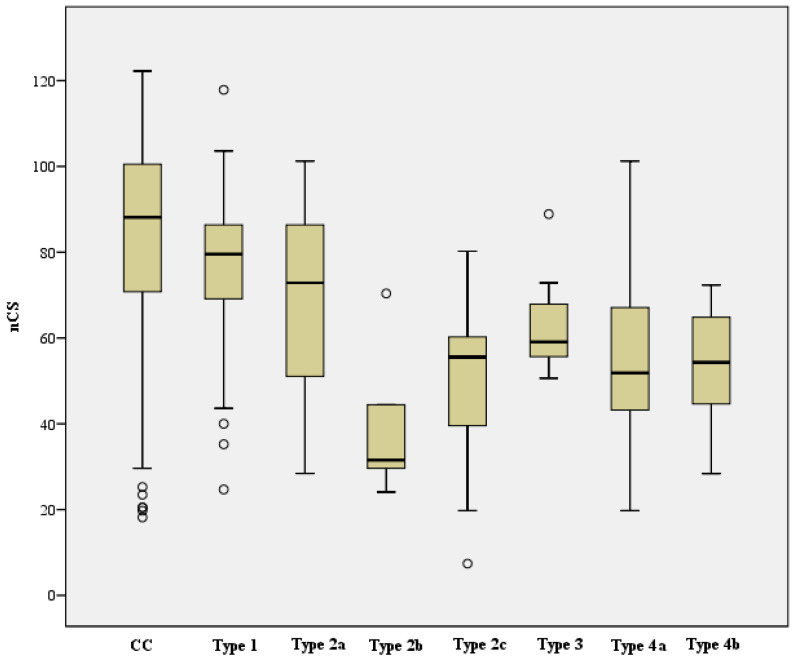
Comparison nCS of collective without complications and complication types (X-axis: study collective, Y-axis: median, IQR, extreme values, outliers).

**Figure 24 jcm-12-02556-f024:**
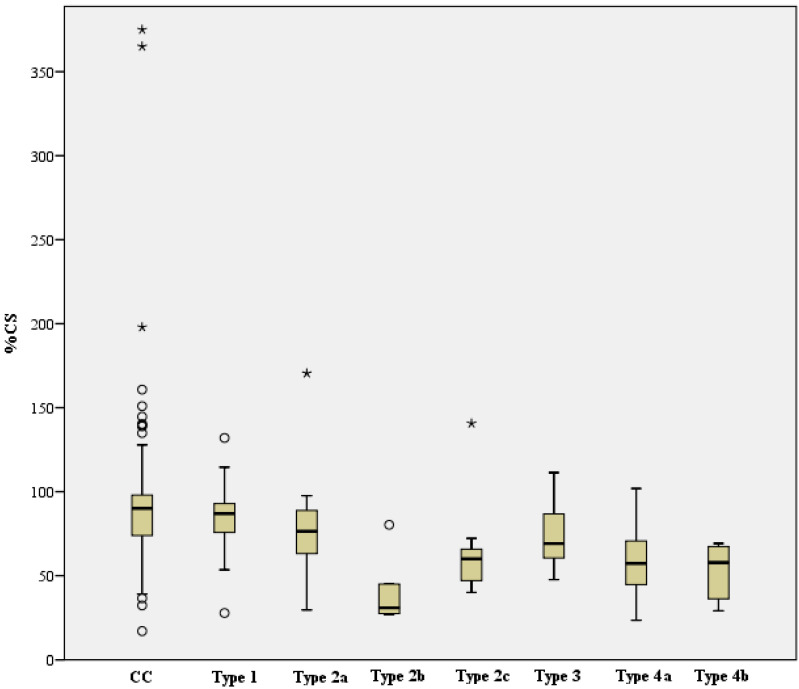
Comparison %CS of collective without complications and complication types (X-axis: study collective, Y-axis: median, IQR, o= extreme values, * = outliers).

**Figure 25 jcm-12-02556-f025:**
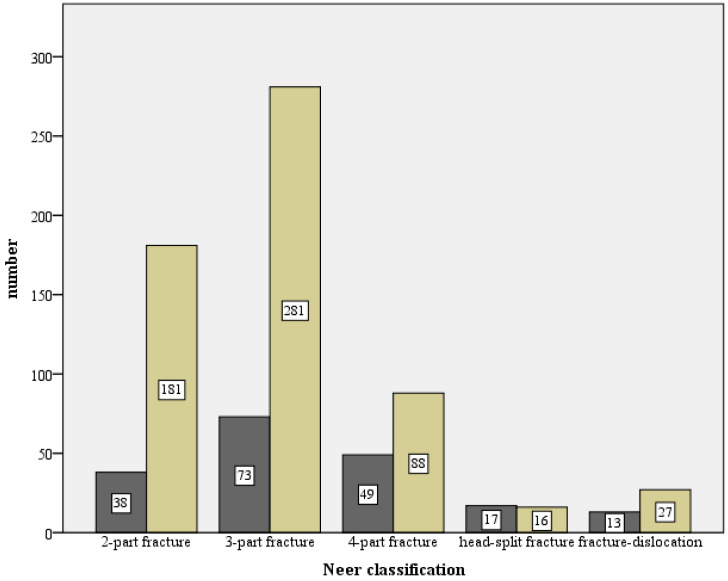
Comparison of the number with complications (dark) and without complications (light) in relation to the respective fracture type, based on the Neer classification (X-axis: Neer classification, Y-axis: number).

**Figure 26 jcm-12-02556-f026:**
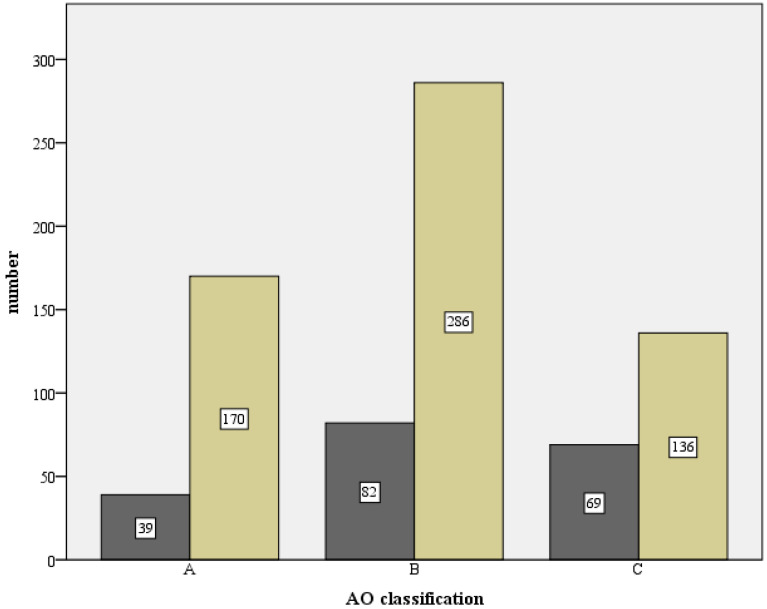
Comparison of the number with complications (dark) and without complications (light) in relation to the respective fracture type, based on the AO classification (X-axis: AO classification, Y-axis: number).

**Table 1 jcm-12-02556-t001:** Classification of complication types following locking plate osteosynthesis of displaced fractures of the proximal Humerus.

Complication Type	Exemplary Images by X-rays or Pattern Images	Definition
1	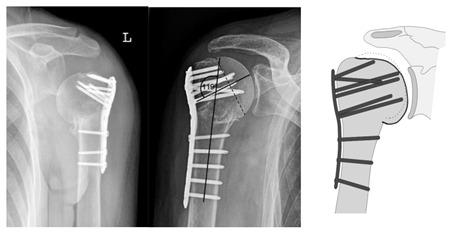	Complication Type 1 is defined by a mild, stable varus or valgus displacement (<20°) of the humeral head without resulting in a screw cutout through the humeral head cortex.
2a	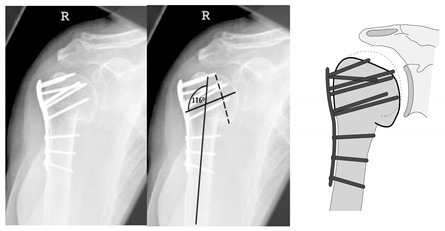	Complication Type 2a is defined by varus displacement (<20°) of the humeral head associated with screw cutout through the humeral head cortex. The head sintering is considered relatively stable and non-progressive.
2b	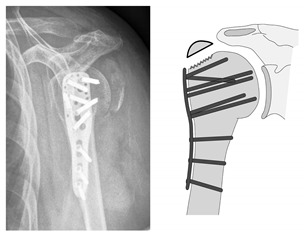	Complication Type 2b is limited to displacement of the greater tuberosity, lesser tuberosity, or both tuberosities.
2c	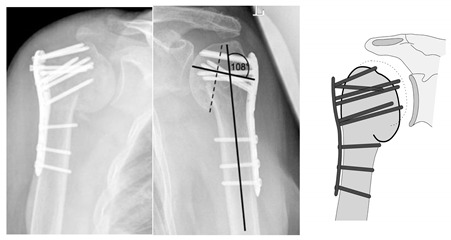	Type 2c is defined by a severe varus displacement (>20°) of the humeral head, which is associated with a screw cutout at the humeral head in parallel to Type 2a. In contrast to Type 2a, this varus displacement is characterized by an unstable osteosynthesis situation and progression.
3	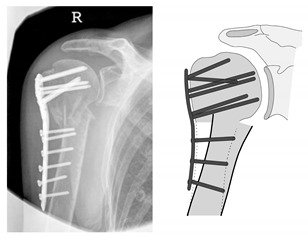	Type 3 describes a displacement of the angular stable plate osteosynthesis with screw cutout/fracture avulsion in the humeral shaft region, while the humeral head position remains intact.
4	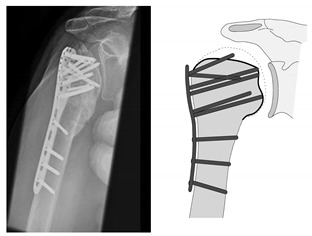	Complication Type 4a is based on the presence of AVN of the humeral head without concomitant destruction of the glenoid articular surface due to an associated screw cutout through the humeral head cortex.
4b	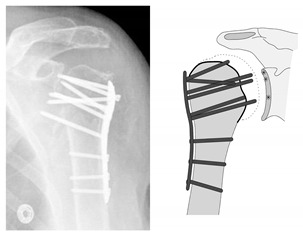	Complication Type 4b is also based on the presence of an AVN of the humeral head combined with a destruction of the glenoid articular surface due to the associated screw cutout through the humeral head cortex.

**Table 2 jcm-12-02556-t002:** Descriptive distribution of the complication types.

	n	% Complications	% Total (n = 787)	Age in Years (Mean, SD)	Distribution Men to Women, Percent Women (%)
Type 1	54	28.0	6.9	69.1 ± 12.1	19/35
Type 2a	41	21.2	5.2	75.1 ± 11.9	11/30
Type 2b	13	6.7	1.6	77.8 ± 7.5	2/11
Type 2c	25	13.0	3.2	71.4 ± 13.1	7/18
Type 3	18	9.3	2.3	82.7 ± 9.1	6/12
Type 4a	34	17.6	4.3	66.4 ± 13.0	12/22
Type 4b	8	4.2	1.0	69.3 ± 14.6	4/4
Total Complications	193	100	24.5	72.1 ± 12.9	61/132 (68.4%)
Total	787			66.5 ± 15.6	255/532 (67.6%)

**Table 3 jcm-12-02556-t003:** Functional outcomes by means of mean and standard deviation of CS, nCS, and %CS of the case cohort; cohort without complications; complication types; and the complication collective in total as well as the respective follow-up of the functional scores in years (mean, standard deviation and median).

	CS (Mean, SD)	nCS (Mean, SD)	%CS (Mean, SD)	Follow-up (Mean, SD; Median)
Case Cohort	68.4 ± 20.3 p. (n = 557)	80.4 ± 23.8 p. (n = 557)	83.9 ± 28.1% (n = 517)	CS/nCS: 4.0 ± 2.7; 3.3 %CS: 3.5 ± 2.5; 2.8
Cohort without complications CC	72.1 ± 19.0 p. (n = 440)	84.7 ± 21.9 p. (n = 440)	87.2 ± 27.9% (n = 411)	CS/nCS: 4.2 ± 2.9; 3.2 %CS: 4.4 ± 2.7; 3.8
Cohort with complications CWC	54.5 ± 19.0 p. (n = 117)	64.5 ± 229 p. (n = 117)	71.2 ± 25.0% (n = 106)	CS/nCS: 3.3 ± 2.5; 2.8 %CS: 3.5 ± 2.5; 2.8
Type 1	64.7 ± 16.7 p. (n = 37)	76.2 ± 20.3 p. (n = 37)	84.5 ± 18.0% (n = 31)	CS/nCS: 3.2 ± 1.9; 3.1 %CS: 3.5 ± 1.8; 3.3
Type 2a	58.7 ± 17.5 p. (n = 29)	70.0 ± 21.6 p. (n = 29)	77.5 ± 24.6% (n = 28)	CS/nCS: 2.7 ± 2.2; 1.9 %CS: 2.8 ± 2.2; 2.1
Type 2b	33.2 ± 13.0 p. (n = 5)	40.0 ± 16.6 p. (n = 5)	42.0 ± 20.2% (n = 5)	CS/nCS/%CS: 1.2 ± 0.8; 0.9
Type 2c	39.3 ± 17.1 p. (n = 9)	48.0 ± 21.1 p. (n = 9)	65.3 ± 28.5% (n = 9)	CS/nCS/%CS: 3.3 ± 2.7; 2.2
Type 3	52.7 ± 9.4 p. (n = 7)	63.7 ± 12.2 p. (n = 7)	74.1 ± 20.5% (n = 6)	CS/nCS: 2.7 ± 2.7; 1.0 %CS: 3.1 ± 2.7; 2.4
Type 4a	46.4 ± 17.2 p. (n = 23)	54.2 ± 20.4 p. (n = 23)	57.3 ± 20.4% (n = 21)	CS/nCS: 4.7 ± 3.2; 3.5 %CS: 5.0 ± 3.3; 4.0
Type 4b	46.3 ± 14.0 p. (n = 7)	53.4 ± 14.9 p. (n = 7)	52.9 ± 15.1% (n = 6)	CS/nCS: 3.6 ± 2.1; 3.3 %CS: 3.8 ± 2.2; 4.0

**Table 4 jcm-12-02556-t004:** Need for revision and distribution of revisions performed depending on the complication type. Depending on the complication type, the dominant procedures are highlighted by the dark-shaded fields.

	Type 1	Type 2a	Type 2b	Type 2c	Type 3	Type 4a	Type 4b	Total
Early-IR (<9 Month.)	5	9 (35%)	3 (25%)	2 (8%)	1 (7%)	13 (39%)	2 (20%)	35
IR	9	4 (15%)	1 (8%)	2 (8%)	0	7 (21%)	0	23
screw replacement/removal	1	13 (50%)	1 (8%)	3 (12%)	0	5 (15%)	0	23
Reosteosynthesis	0	0	3 (25%)	4 (16%)	11 (73%)	0	0	18
Resection	0	0	0	2 (8%)	0	0	1 (10%)	3
intramedullary nailing	0	0	0	1 (4%)	2 (13%)	0	0	3
Hemiarthroplasty	0	0	2 (17%)	4 (16%)	1 (7%)	5 (15%)	0	12
Reverse Arthroplasty	0	0	2 (17%)	7 (28%)	0	3 (10%)	7 (70%)	19
Total	15	26	12	25	15	33	10	136
No Revision	39	17	5	4	4	6	0	75
Revision rate	28%	59%	62%	84%	78%	82%	100%	61%

**Table 5 jcm-12-02556-t005:** Percentage frequencies of complication development in association with the fracture type of the Neer classification.

Neer- Classification	2-Part Fracture	3-Part Fracture	4-Part Fracture	Head-Split-Fracture	Fracture-Dislocation
% Complication	17.4	20.6	35.8	51.5	32.5

**Table 6 jcm-12-02556-t006:** Distribution of complication types among the individual fracture types of the Neer classification and AO classification. The fields with a dark background reflect the dominant fracture group(s) of the respective complication types (with associated percentage value).

	Type 1	Type 2a	Type 2b	Type 2c	Type 3	Type 4a	Type 4b
2-part fracture	12	9	0	8 (35%)	7 (39%)	2	0
3-part fracture	24 (44%)	19 (46%)	5	8 (35%)	9 (50%)	7	1
4-part fracture	12	9	6 (46%)	5	0	14 (41%)	3 (43%)
Head-Split-Fractur	2	2	1	0	2	7 (21%)	3 (43%)
Fracture-Dislocation	4	2	1	2	0	4	0
A2	7	2	0	2	2	1	0
A3	6	6	0	6 (26%)	6 (33%)	1	0
B1	12 (22%)	12 (29%)	2	2	1	2	1
B2	14 (26%)	6	3	7 (30%)	7 (39%)	5	0
B3	2	2	0	1	0	3	0
C1	2	5	2	1	0	1	0
C2	10	6	5 (38%)	3	1	16 (47%)	5 (71%)
C3	1	2	1	1	1	5	1
No X-Rays	0	0	0	2	0	0	1

## Data Availability

Due to privacy or ethical restrictions there is no additional data.

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
