# Peer review of "It Is Always the Same—A Complication Classification following Angular Stable Plating of Proximal Humeral Fractures"

_jcm, 2023, doi:10.3390/jcm12072556_

Round 1

Reviewer 1 Report

line 1-76: Well written introduction section. However, the rationale or justification of the study with the knowledge gap should be highlighted clearly rather than writing extensively about the known facts. 

line 80-82. Please re-write the sentence. This looks like statement rather than objective. 

line 85-86: Can you provide some information on how you calculated the sample size required? The study design needs to be changed, as it seems more of a retrospective case series than Cohort study. Please get help from statistician and provide justification.

line 101-110: Excellent representation of the account of every patient. Authors must be credited for honest representation of the data. 

line 113-119: Please provide citation whenever necessary. 

line 133-159: Excellent. Clearly written statistical analysis section. 

line 160-167: please provide details of ethical considerations. Ethical review board review status with reference number. 

Result section seems to have mixed with methodology section. I acknowledge that it was written this way to make clear understanding for the readers. However, it would have been better if you explain how you planned to classify the complications in the methodology section. Result section should mainly have the outcomes. 

Please avoid the repetition of the findings. For e.g., those mentioned clearly in the graphs, charts, or table, need not be repeated in the sentences. Please correct throughout the result section accordingly. 

A flow diagram of complication types and their definition would be clearer to understand rather than table. 

Author Response

Dear reviewer,

Thank you very much for the thorough revision of our manuscript, we are very thankful for the constructive input and your comments.

In the following, we illustrate how we completely addressed the reviewers’ concerns by giving detailed answers to the reviewers’ comments and stating the specific changes made to the manuscript. Our responses are written in italic font, changes to the manuscript are underlined, deleted sections crossed out.

We revised our manuscript according to the proposed modifications and think that this substantially improved our paper.

  1. line 1-76: Well written introduction section. However, the rationale or justification of the study with the knowledge gap should be highlighted clearly rather than writing extensively about the known facts. 

Thank you very much for your comment. We agree with you that the complication rates are known to be high and therefore explained in the introduction section. Unfortunately there is no clinical work so far, analyzing iterating complications following angular stable plate osteosynthesis of displaced proximal humeral fractures and extrapolate recommendations on complication management.   We have presented this in detail in the discussion.

  1. line 80-82. Please re-write the sentence. This looks like statement rather than objective. 

Thank you for this comment. We rewrote the sentence

Furthermore, we tried to investigate the impact of these complications on clinical outcome, relation to fracture morphology as well as potential revision strategies. 

  1. line 85-86: Can you provide some information on how you calculated the sample size required? The study design needs to be changed, as it seems more of a retrospective case series than Cohort study. Please get help from statistician and provide justification.

Thank you very much for pointing this out. All consecutive patients were retrospectively followed up and evaluated but prospectively included into a database. In consultation with the Institute for Statistics of the LMU Munich, a retrospective cohort study was chosen as study design. Since the patients were consecutively included in a maximum care clinic and a high case number, no power analysis was calculated before the beginning of this study.

  1. line 101-110: Excellent representation of the account of every patient. Authors must be credited for honest representation of the data. 

Thank you very much for the compliment, we are very honoured by it.

  1. line 113-119: Please provide citation whenever necessary. 

Thank you for the comment, we have added the quotes.  Sproul RC, Iyengar JJ, Devcic Z, Feeley BT. A systematic review of locking plate fixation of proximal humerus fractures. Injury. 2011;42:408-13.

  1. line 133-159: Excellent. Clearly written statistical analysis section. 

Thank you very much for the comment, we are very honoured.

  1. line 160-167: please provide details of ethical considerations. Ethical review board review status with reference number. 

Thank you for the comment. The details of the ethical considerations are pointed out in line 682-684: The study was conducted in accordance with the Declaration of Helsinki, and approved by the Ethics Committee of LMU (protocol code 156-12 date of approval 8 of May 2012).

  1. Result section seems to have mixed with methodology section. I acknowledge that it was written this way to make clear understanding for the readers. However, it would have been better if you explain how you planned to classify the complications in the methodology section. Result section should mainly have the outcomes. 

Thank you for the explanation. We had the same thought and debated it intensively in the working group. To improve the understanding of the reader we have chosen this way as the complication types were established after prior publications and first analysis of the typical patterns.

  1. Please avoid the repetition of the findings. For e.g., those mentioned clearly in the graphs, charts, or table, need not be repeated in the sentences. Please correct throughout the result section accordingly. 

Due to the better comprehensibility without studying the tables, we partly discuss the results already listed in the tables again and thus partly point out the most important results. 

  1. A flow diagram of complication types and their definition would be clearer to understand rather than table. 

Thank you very much for the advice, we had already thought about this in detail, but a flow chart with practical use was not possible due to deadlines. Therefore we decided against it but are planning to publish a clinical flowchart including complication patterns and revision strategies once our proposal is accepted in the shoulder and elbow community.

Reviewer 2 Report

1) First of all, the author should explain why osteosynthesis-related complication divied into the following classification systems, initially. No part explains why it was divided based on varus/valgus displacement, screw cutout. Furthemore, why did you set the standard for severity of varus displacement to 20 degrees in this classification system ?

2) Was the rotator cuff repair performed when the rotator cuff tear was accompanied during the fracture fixation?

3) Were no difference in bone mineral density (BMD) observed in this classification system ? In case of type III, I think lower BMD (osteoporotic pateints), the more likely it will be.

4) In conclusion, for actual clinical applications, what can be gained from this classification system? Do the surgeons have to do varus reduction as anatomical position during surgical procedures ? Overall, it is well-organized and seems to be a reasonable. However, the questions remain that it provides practical guidelines, which need to be implemented.

Minor issues

1. The simple introduction should be included before “The aim of the study” in Abstract section, which may give a easy-to-read for readers.

2. There are many paragraphs in the Introduction section. I hope that the paragraphs are combined.

3. 2.1 selection criteria section, page 3, line 106-107, CS (Constant score) and nCS (normalized constant score) à Constant score (CS) and normalized constant score (nCS)

4. Page 3, line 108 : 4,0±2,7 years à 4.0±2.7 years

5. Same references were duplicated : (Ref 1 and Ref 4) (Ref 9 and 11) and reference form is not matched with the form of Journal of Clinical Medicine.

Author Response

Dear reviewer,

Thank you very much for the thorough revision of our manuscript, we are very thankful for the constructive input and your comments.

In the following, we illustrate how we completely addressed the reviewers’ concerns by giving detailed answers to the reviewers’ comments and stating the specific changes made to the manuscript. Our responses are written in italic font, changes to the manuscript are underlined, deleted sections crossed out.

We revised our manuscript according to the proposed modifications and think that this substantially improved our paper.

Major issues

  1. First of all, the author should explain why osteosynthesis-related complication divied into the following classification systems, initially. No part explains why it was divided based on varus/valgus displacement, screw cutout. Furthemore, why did you set the standard for severity of varus displacement to 20 degrees in this classification system ?

Thank you very much for the evaluation of our work. We appreciate your comment. We describe the complication types in accordance with Fleischhacker et al. (DOI: 10.1016/j.injury.2020.09.003). This 2020 published study demonstrated a relation of the humeral head-shaft angle to functional outcome following ORIF of proximal humeral fractures. Screw cutout occurs typically in cases with severe (>20°) varus displacement.

Osteosynthesis-related complications were divided into the following classification system in accordance to the 2020 published study by Fleischhacker et. al, which demonstrated the relation of varus displacement to functional outcome following ORIF of proximal humeral fractures. Functional clinical outcome is inferior in cases with   humeral head- shaft angle displacement of 10 to 20 ° varus in comparison to an anatomical head- shaft angle of 135 °. A varus malposition/displacement of> 20° resulted in even worse functional outcome as well as screw cutout.

  1. Was the rotator cuff repair performed when the rotator cuff tear was accompanied during the fracture fixation?

Thank you for this question. The rotator cuff was intraoperatively evaluated and in cases of ruptures, if possible, fixated to the plate with sutures.

The rotator cuff was intraoperatively evaluated and in cases of ruptures, if possible, fix-ated to the plate with sutures during the operative procedure.

  1. Were no difference in bone mineral density (BMD) observed in this classification system ? In case of type III, I think lower BMD (osteoporotic pateints), the more likely it will be.

Thank you for pointing this out. Unfortunately the bone quality was not always recorded in a standardized way (DXA). Therefore a statistical correlation was not calculated in this study but is topic of ongoing research of our group.

  1. In conclusion, for actual clinical applications, what can be gained from this classification system? Do the surgeons have to do varus reduction as anatomical position during surgical procedures ? Overall, it is well-organized and seems to be a reasonable. However, the questions remain that it provides practical guidelines, which need to be implemented.

Thank you for this important remark. In our opinion this study provides practical guidelines for surgeons how to classify typical iterating osteosynthesis related complications following angular stable plate osteosynthesis of displaced proximal humeral fractures  and offers recommendations on complication management. Type 1 should be treated by skillful neglect, Type 2a with early implant removal and/or screw replacement/screw removal, Type 2b, Type 2c, 3 and 4 demand for revision osteosynthesis or arthroplasty.

Minor issues

  1. The simple introduction should be included before “The aim of the study” in Abstract section, which may give a easy-to-read for readers.

Thank you for this useful comment. We added a simple introduction.

Abstract Introduction: The aim of this study was to create a novel complication

classification for iterating osteosynthesis related complications following angular stable

plate osteosynthesis of displaced proximal humeral fractures 

  1. There are many paragraphs in the Introduction section. I hope that the paragraphs are combined.

Thank you for your helpful comment. We have combined some paragraphs.

  1. 1 selection criteria section, page 3, line 106-107, CS (Constant score) and nCS (normalized constant score) à Constant score (CS) and normalized constant score (nCS)

Thank you for reporting this typo. We have corrected it.

  1. Page 3, line 108 : 4,0±2,7 years à 4.0±2.7 years

Thank you for reporting typo. We have corrected it.

  1. Same references were duplicated : (Ref 1 and Ref 4) (Ref 9 and 11) and reference form is not matched with the form of Journal of Clinical Medicine.

Thank you for reporting this transmission error. We have corrected it.

Reviewer 3 Report

Dear authors it a good article. Written well but need many changes in the manuscript before final shape. Please go through the comments file attached and do the required changes.

Best regards

Author Response

Dear reviewer 2,

Thank you very much for the thorough revision of our manuscript, we are very thankful for the constructive input and your comments.

In the following, we illustrate how we completely addressed the reviewers’ concerns by giving detailed answers to the reviewers’ comments and stating the specific changes made to the manuscript. Our responses are written in italic font, changes to the manuscript are underlined, deleted sections crossed out.

We revised our manuscript according to the proposed modifications and think that this substantially improved our paper.

  1. Abstract – Is ok

Thank you.

  1. Methodology- A collective of 1031 patients with 1047 displaced proximal humerus fractures were included in this retrospective cohort study at the Musculoskeletal University Center Munich of the Ludwig-Maximilians-University Munich. Please clarify the total numbers of subjects it is 1031 or 1047

Thank you for drawing attention to possible problems of understanding. In fact, 1031 patients with 1047 displaced proximal humerus fractures could be included from February 2002 and December 2014. In some cases the patients sustained fractures of both sides at different time points.  We added this to the method section.

In some cases the patients sustained fractures of both sides at different time points. 

  1. Please clearly write the inclusion criteria for your study.

Thank you for pointing this out, we have added an inclusion paragraph.

The inclusion criteria for this study were consecutive patients (>18 years) treated by angular stable plate osteosynthesis of a displaced proximal humeral fracture. The surgical indication for ORIF was made in all patients based on Neer's criteria (angulation >45°, fracture displacement >1cm).

  1. The materials and methods should be described with sufficient details to allow others to replicate and build on the published results. Please note that the publication of your manuscript implicates that you must make all materials, data, computer code, and protocols associated with the publication available to readers. Please disclose at the submission stage any restrictions on the availability of materials or information. New methods and protocols should be described in detail while well-established methods can be briefly described and appropriately cited.

Thank you very much for your useful comment. In our opinion, methods and material were described in detail as well as the complete results that were subject of this study are mentioned in the article. Furthermore, no new methods were used in this study, but methods established from other studies reporting on clinical outcome and complications which were cited appropriately.

  1. Total numbers of included subjects were 557 and radiological assessment was done for 787- Please clarify it.

Thank you for this comment. Clinical total follow-up regarding Constant score was available in 557 (53.2%) cases, with a mean follow-up of 4.0±2.7 years and median of 3.3 years. More patients could be completely followed-up radiologically without a complete clinical outcome. In total, 787 cases (75.2%) were available for radiological outcome/complication analysis with different follow-up time. The mean follow-up was 7.3±14.0 months, with a median of 2.8 months. We added the information to be more precise.

Clinical total follow-up regarding Constant score was available in 557 (53.2%) cases, with a mean follow-up of 4.0±2.7 years and median of 3.3 years and the mean age of the 557 cases was 65.3±14.5 years. More patients could be completely followed-up radiologically without a complete clinical outcome. In total, 787 cases (75.2%) were available for radiological outcome/complication analysis with different follow-up time. The mean follow-up was 7.3±14.0 months, with a median of 2.8 months. The mean age of the 787 patients was 66.5±15.6 years, and the gender distribution showed 532 (67.6%) women and 255 (32.4%) men.

  1. In Result section- All p values must be written in Italic- p, that is standard.

Thank you for pointing out this error. We have corrected it.

  1. Discussion- Line 643 - Individual adaptation to the patient's needs and anatomic conditions should be made. Please explain what do you mean- Is this you want from future studies?

Thank you for pointing this out, due to the misunderstanding the sentence has been deleted.

  1. Limitations- from line 657-660- Authors should discuss the results and how they can be interpreted from the perspective of previous studies and of the working hypotheses. The findings and their implications should be discussed in the broadest context possible. Future research directions may also be highlighted. Please remove it.

Thank you for pointing this out. The lines have been deleted. 

  1. Abbreviations table can be added or next to every new abbreviations write it full name in the text.

  1. Thank you for pointing this out. We have added abbreviations with full names within the text.

  1. References- Year of the publication should me made Bold.

Thank you for your comment, we have revised the references.

Round 2

Reviewer 2 Report

No more comment

Reviewer 3 Report

Dear athors it is ok now you have did the required changes. now the manuscript look much better. 

All the best & regards.